# Universal Causal Inference in a Topos

**Sridhar Mahadevan**[*]
Adobe Research
San Jose, CA
smahadev@adobe.com

## Abstract

In this paper, we explore the universal properties underlying causal infer-
ence by formulating it in terms of a *topos*. More concretely, we introduce
topos causal models (TCMs), a strict generalization of the popular structural
causal models (SCMs). A topos category has several properties that make
it attractive: a general theory for how to combine local functions that
define "independent causal mechanisms" into a consistent global function
building on the theory of sheaves in a topos; a generic way to define causal
interventions using a subobject classifier in a topos category; and finally,
an internal logical language for causal and counterfactual reasoning that
emerges from the topos itself. A striking characteristic of subobject classi-
fiers is that they induce an intuitionistic logic, whose semantics is based
on the partially ordered lattice of subobjects. We show that the underlying
subobject classifier for causal inference is not Boolean in general, but forms
a Heyting algebra. We define the internal Mitchell-Bénabou language, a
typed local set theory, associated with causal models, and its associated
Kripke-Joyal intuitionistic semantics. We prove a universal property of
TCM, namely that any causal functor mapping decomposable structure to
probabilistic semantics factors uniquely through a TCM representation.

## 1 Introduction

In recent years, there has been significant interest in categorical models of causality, based on
symmetric monoidal categories [Fong, 2012, Fritz and Klingler, 2023, Cho and Jacobs, 2019,
Jacobs et al., 2018], as well as simplicial sets and higher-order categories [Mahadevan, 2023].
Markov categories [Fritz, 2020] define a broad unifying framework for probabilistic inference
and statistics using symmetric monoidal categories, where each object is additionally
equipped with a comonoidal "copy-delete" operation. It enables carrying out rigorous
proofs using an elegant string diagrammatic language [Selinger, 2010]. Any causal model
based on graphs [Pearl, 2009, Forré and Mooij, 2017, Spirtes et al., 2000] or other algebraic
formalisms, such as integer-valued multisets [Studeny, 2010], can be translated into a string
diagram over a symmetric monoidal category, or a simplicial set. Operations on causal
models, such as interventions, can be modeled as functors on the objects of the associated
symmetric monoidal category or simplicial set. Categorical approaches to causality also
extend to the *potential outcomes* counterfactual framework [Imbens and Rubin, 2015].

Categorical approaches fundamentally differ from past work in causality in their focus on
the elucidation of universal properties. In our previous work [Mahadevan, 2023, 2025c], we
introduced the framework of *universal causality* based on the notion of universal properties

---

[*]Academic affiliation: Research Professor, University of Massachusetts, Amherst; See webpage at
https://people.cs.umass.edu/~mahadeva/Site/About_Me.html

39th Conference on Neural Information Processing Systems (NeurIPS 2025).

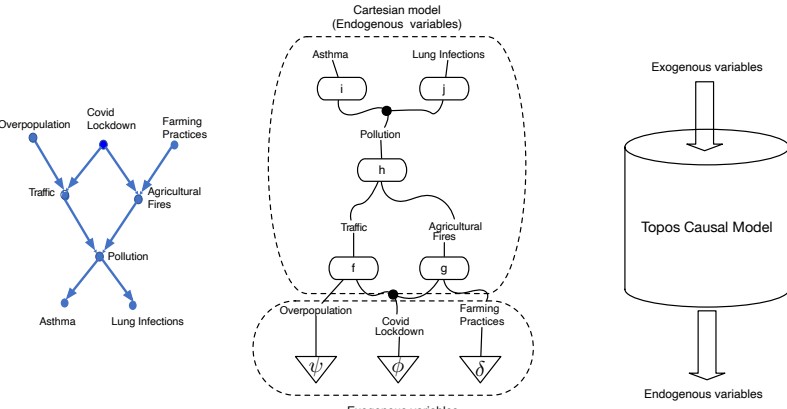

Figure 1: Topos causal models (TCMs) are defined as a category $C_{\mathcal{TCM}}$ whose objects $c \in C_{\mathcal{TCM}}$ are causal models, and whose arrows $C_{\mathcal{TCM}}(c, c')$ are commutative diagrams between models $c$ and $c'$. A specific object $c$ defining a model can be conceptualized as a DAG (left, where information flows from top to bottom), or a string diagram in a Markov category (middle, where information flows from bottom to top), or in terms of its induced unique "blackbox" function mapping exogenous variables to endogenous variables (right).

in category theory [Riehl, 2017]: a causal property is universal if it can be defined in terms of an *initial* or *final* object in a category of causal diagrams, or in terms of a *causal representable functor* using the Yoneda Lemma. For example, a structural causal model (SCM) [Pearl, 2009] is defined as a (deterministic) mapping from a collection of exogenous variables into a collection of endogenous variables, derived by "collating" local functions that serve as independent causal mechanisms [Galles and Pearl, 1988, Parascandolo et al., 2017]. However, SCMs can be further analyzed in terms of their universal properties, such as categorical product, coproduct, limits and colimits, equalizers and coequalizers etc. These latter properties can be shown formally to be initial or final objects in a category of diagrams [Riehl, 2017], or as representable functors through the Yoneda Lemma [MacLane, 1971].

Our main contribution in this paper is to present a *topos-theoretic* view of causality, and in particular, introduce topos causal models (TCMs) that strictly generalize structural causal models (SCMs) [Pearl, 2009]. A topos is a type of category [MacLane, 1971], which is particularly well-suited to modeling operations that are "set-like" [MacLane and leke Moerdijk, 1994]. It also features an internal logical language [Goldblatt, 2006]. We claim that a topos provides three universal properties that make it natural as a category to do causal inference in: it provides a general theory for how to combine local functions, which can be viewed as "independent causal mechanisms" [Parascandolo et al., 2017], into a consistent global function building on the theory of sheaves in a topos [Mac Lane and Moerdijk, 1992]. It enables a generic way to define causal interventions using a subobject classifier in a topos category [Johnstone, 2014]. Finally, it gives an internal logical language for causal and counterfactual reasoning [Bell, 1988].

As Figure 1 illustrates, the objects in a TCM category can be conceptualized in multiple ways. First, each object can be a causal graphical model [Pearl, 1989, Spirtes et al., 2000]. Each object can also be a functor: for example, directed graphs form a topos functor category [Vigna, 2003]. TCMs can also be defined in terms of *string diagrams* in a symmetric monoidal *Markov category* [Fritz, 2020], where we restrict ourselves to the Markov subcategory defined through deterministic morphisms. For example, the arrow $h$ : Traffic⊗Agricultural Fires → Pollution defines a deterministic mapping specifying the two potential causes of Pollution. For exogenous variables, the arrow $\psi : I \rightarrow$ Overpopulation defines the marginal distribution on Overpopulation, where $I$ is the terminal object in the Markov category. Finally, we can view a TCM object as a "blackbox" function that maps some collection of exogenous variables (e.g., "Overpopulation", or "Farming Practices" into some set of endogenous variables, e.g., "Asthma" or "Pollution").

## 2 Principles of Universal Causality

We give a brief overview of the fundamentals of universal causality (UC) [Mahadevan, 2023, 2025c] before delving into the specific details of the TCM framework. As with other work in categorical causality [Fong, 2012, Jacobs et al., 2018, Fritz and Klingler, 2023], UC uses category theory [MacLane, 1971] to define causality. A category $C$ is a collection of abstract *objects* $c \in C$. Anything technically can count as an object, from a variable in a causal model to an entire model itself. Each category $C$ is additionally specified by a set of arrows or morphisms $C(c, d)$ between each pair of objects $c$ and $d$. There is an identity arrow $1_c \in C(c, c)$. Arrows compose in the obvious way, inducing a function $C(c, d) \times C(d, e) \rightarrow C(c, e)$. An *initial object* $c$ in category $C$ defined as one inducing a unique arrow from $c$ to every object in category $C$. A *terminal* object, usually denoted by $\mathbf{1}$, is one that defines a unique arrow from every object $c$ in category $C$ into $\mathbf{1}$. An object $c$ is isomorphic to another object $d$, denoted $c \simeq d$, if two arrows $f : c \rightarrow d$ and $g : d \rightarrow c$ exist, such that $g \circ f = 1_c$, and $f \circ g = 1_d$. A *functor* $F : C \rightarrow \mathcal{D}$ between two categories $C$ and $\mathcal{D}$ is specified by an *object function* mapping each $c \in C$ to $Fc \in \mathcal{D}$, and an *arrows function* mapping each arrow $f \in C(c, d)$ to $Ff \in \mathcal{D}(Fc, Fd)$. Functors come in two varieties – covariant and contravariant – the latter acts on the domain category by reversing the arrows. Given any two functors $F : C \rightarrow D$ and $G : C \rightarrow D$ between the same pair of categories, we can define a mapping between $F$ and $G$ that is referred to as a natural transformation. These are defined through a collection of mappings, one for each object $c$ of $C$, thereby defining a morphism in $D$ for each object in $C$.

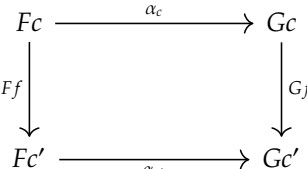

### 2.1 Yoneda Lemma and the Causal Reproducing Property

UC rests on the Yoneda Lemma – any object in a category can be defined by the interactions it makes with other objects (upto isomorphism). In the setting of causal inference, it means that objects in a TCM category can be ascribed "meaning" through studying the arrows of the category, without having to "look inside" the object. The Yoneda Lemma states that the set of all morphisms into an object $d$ in a category $C$, sometimes denoted as $\mathbf{Hom}_C(-, d)$, or as $C(-, d)$, denoted as the *presheaf*, is sufficient to define $d$ up to isomorphism. The category of all presheaves forms a *category of functors*, and is denoted $\hat{C} = \mathbf{Set}^{C^{op}}$. This category forms a topos, and will be fundamental to the TCM framework.

**Lemma 1.** *[MacLane, 1971, Riehl, 2017]* **Yoneda lemma***: For any functor $F : C \rightarrow \mathbf{Set}$, whose domain category $C$ is "locally small" (meaning that the collection of morphisms between each pair of objects forms a set), and any object $c$ in $C$, there is a bijection $Hom(C(-, c), F) \simeq Fc$ that associates a natural transformation $\alpha : C(-, c) \Rightarrow F$ to the element $\alpha_c(1_c) \in Fc$. This correspondence is natural in both $c$ and $F$.*

**Definition 1.** *[Riehl, 2017] A* **universal property** *of an object $c \in C$ in a category $C$ is expressed by a representable functor $F$ together with a universal element $x \in Fc$ that defines a natural isomorphism $C(-, c) \simeq F$. The collection of morphisms $C(-, c)$ into an object $c$ is called the* **presheaf***, and from the Yoneda Lemma, forms a universal representation of the object.*

### 2.2 Diagrams and Universal Constructions

A key distinguishing feature of category theory is the use of diagrammatic reasoning. However, diagrams are also viewed more abstractly as functors mapping from some indexing category to the actual category. Diagrams are useful in understanding universal constructions, such as limits and colimits. Briefly, a diagram $F : \mathcal{J} \rightarrow C$ is a functor $F$ from some finite category $\mathcal{J}$ into a category of interest, $C$. For example, $\mathcal{J} = \bullet \rightarrow \bullet \leftarrow \bullet$ is an example of a "pullback" diagram. Here the $\bullet$ refer to abstract objects that are mapped into concrete objects in $C$ by the functor $F$. What we want to know whether a particular diagram

*F* or an entire class of diagrams is "solvable". What this means is whether its limit or colimit exists, that is, is the category *complete* or *co-complete*? For any object $c \in C$ and any category *J*, the *constant functor* $c : J \rightarrow C$ maps every object *j* of *J* to *c* and every morphism *f* in *J* to the identity morphisms $1_c$. We can define a constant functor embedding as the collection of constant functors $\Delta : C \rightarrow C^J$ that send each object *c* in *C* to the constant functor at *c* and each morphism $f : c \rightarrow c'$ to the constant natural transformation, that is, the natural transformation whose every component is defined to be the morphism *f*.

**Definition 2.** *[Riehl, 2017] A* **cone over** *a diagram $F : J \rightarrow C$ with the* **summit** *or* **apex** $c \in C$ *is a natural transformation $\lambda : c \Rightarrow F$ whose domain is the constant functor at c. The components $(\lambda_j : c \rightarrow Fj)_{j \in J}$ of the natural transformation can be viewed as its* **legs**. *Dually, a* **cone under** *F with* **nadir** *c is a natural transformation $\lambda : F \Rightarrow c$ whose legs are the components $(\lambda_j : F_j \rightarrow c)_{j \in J}$.*

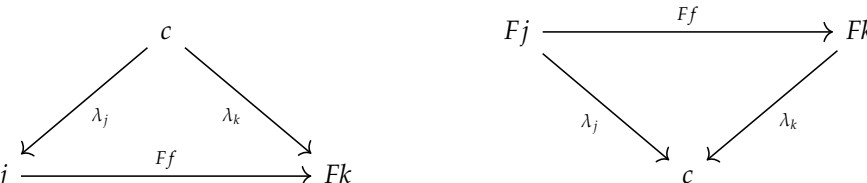

Cones under a diagram are referred to usually as *cocones*. Using the concept of cones and cocones, we can now formally define the concept of limits and colimits more precisely.

**Definition 3.** *[Riehl, 2017] For any diagram $F : J \rightarrow C$, there is a functor $Cone(-, F) : C^{op} \rightarrow$ **Set**, which sends $c \in C$ to the set of cones over F with apex c. Using the Yoneda Lemma, a* **limit** *of F is defined as an object $\lim F \in C$ together with a natural transformation $\lambda : \lim F \rightarrow F$, which can be called the* **universal cone** *defining the natural isomorphism $C(-, \lim F) \simeq Cone(-, F)$. Dually, for colimits, we can define a functor $Cone(F, -) : C \rightarrow$ **Set** that maps object $c \in C$ to the set of cones under F with nadir c. A* **colimit** *of F is a representation for $Cone(F, -)$. Once again, using the Yoneda Lemma, a colimit is defined by an object $ColimF \in C$ together with a natural transformation $\lambda : F \rightarrow colimF$, which defines the* **colimit cone** *as the natural isomorphism $C(colimF, -) \simeq Cone(F, -)$.*

Limit and colimits of diagrams over arbitrary categories can often be reduced to the case of their corresponding diagram properties over sets.

## 2.3 The Universality of Diagrams and the Causal Reproducing Property

We state two key results that underly UC [Mahadevan, 2023]. While both these results follow directly from basic theorems in category theory, their significance for causal inference is what makes them particularly noteworthy. The first result pertains to the notion of diagrams as functors, and shows that for the functor category of presheaves, which is a universal representation of causal inference, every presheaf object can be represented as a colimit of representables through the Yoneda Lemma. This result can be seen as a generalization of the very simple result in set theory that each set is a union of one element sets. The second result is the causal reproducing property, which shows that the set of all causal effects between two objects is computable from the presheaf functor objects defined by them. Both these results are abstract, and apply to any category representation of a causal model.

**Theorem 1.** *[MacLane and leke Moerdijk, 1994, Mahadevan, 2023]* **Universality of Diagrams in UC**: *In the functor category of presheaves $\mathbf{Set}^{C^{op}}$, every functor object F is the colimit of a diagram of representable objects, in a canonical way.*

To explain the significance of this result for causal inference, note that UC represents causal diagrams as functors from an indexing category of diagrams to an actual causal model. The theorem above tells us that every presheaf object can be represented as a colimit of (simple) representable objects, namely functor objects of the form $\mathbf{Hom}_C(-, c)$.

**Theorem 2.** *[MacLane, 1971, Mahadevan, 2023]* **Causal Reproducing Property:** *All causal influences between any two objects c and d can be derived from its presheaf functor objects, namely*

$$\mathbf{Hom}_C(c, d) \simeq \mathbf{Nat}(\mathbf{Hom}_C(-, c), \mathbf{Hom}_C(-, d))$$

Any causal influence of an object $c$ upon any other object $d$ can be represented as a natural transformation (a morphism) between two functor objects in the presheaf category $\hat{C}$.

## 3  Topos Causal Models

A topos [Johnstone, 2014] is a "set-like" category which generalizes all common operations on sets. Thus, the generalization of subset is a subobject classifier in a topos. To help build some intuition, consider how to define subsets without "looking inside" a set. Essentially, a subset $S$ of some larger set $T$ can be viewed as a "monic arrow" (an injective function). Monic arrows generalize injective functions.

**Definition 4.** *[Goldblatt, 2006] An arrow $f : a \rightarrow b$ in a category $C$ is called **monic** if given any parallel arrows $g, h : c \rightrightarrows a$, the equality $f \circ g = f \circ h$ implies that $g = h$, namely $f$ is "left-cancellable".*

Our approach builds on this abstraction to define a category $C_{\mathcal{TCM}}$ whose objects are causal models, such as SCMs or Markov categories, and a submodel $M_x$ of an SCM $M$ is simply a monic arrow $f_x : M_x \hookrightarrow M$.

**Definition 5.** *A category $C$ has **binary products** if for every pair of objects, $c$ and $d$, there exists a third object, $e \simeq c \times d$, along with two projection arrows, $p_1 : e \rightarrow c$ and $p_2 : e \rightarrow d$, such that for any other object $a$ and arrows $f : a \rightarrow c$ and $g : a \rightarrow d$, there exists a unique morphism $u : a \rightarrow e$ satisfying $p_1 \circ u = f$ and $p_2 \circ u = g$.*

**Definition 6.** *[MacLane and Ieke Moerdijk, 1994] A category $C$ with binary products has **exponential objects** if for each pair of objects $c, d$ in $C$, there exists an object $c^d$ that defines the following bijection:*

$$C(e \times d, c) \simeq C(e, c^d)$$

**Definition 7.** *[MacLane and Ieke Moerdijk, 1994] A category $C$ is **Cartesian closed** if it has binary products, a terminal object **1**, and exponential objects.*

**Definition 8.** *[MacLane and Ieke Moerdijk, 1994] In a category $C$ with finite limits, a **subobject classifier** is a $C$-object $\Omega$, and a $C$-arrow **true** : $\mathbf{1} \rightarrow \Omega$, such that to every monic arrow $S \hookrightarrow X$ in $C$, there is a unique arrow $\phi$ that forms the following pullback square:*

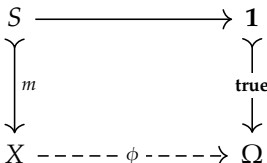

This commutative diagram enforces the constraint that every monic arrow $m$ (i.e., every $1 - 1$ function) that maps a subobject $S$ to an object $X$ must be characterizable in terms of a "pullback", a particular type of universal property that is a special type of a limit. In the special case of the category of sets, subobject classifiers are defined through the characteristic (Boolean-valued) function $\phi$ that defines subsets. In general, as we show below, the subobject classifier $\Omega$ for causal models is not Boolean-valued, and requires using intuitionistic logic through a Heyting algebra. This definition can be rephrased as saying that the subobject functor is representable. In other words, a subobject of a causal model $X$ in category $C_{\mathcal{TCM}}$ is an equivalence class of monic arrows $m : S \hookrightarrow X$.

**Definition 9.** *MacLane and Ieke Moerdijk [1994] An **elementary topos** is a category $C$ that is Cartesian closed and has a subobject classifier.*

For example, the category of sets forms a topos. Limits exist because one can define Cartesian products of sets, and colimits correspond to forming set unions. Exponential objects correspond to the set of all functions between two sets. Finally, the subobject classifier is simply the subset function, which induces a boolean-valued characteristic function.

**Definition 10.** *The category $C_{\mathcal{TCM}}$ of topos causal models is defined as a collection of objects $c \in C_{\mathcal{TCM}}$, each of which is a triple $\langle U, V, F \rangle$ where $V = \{V_1, \ldots, V_n\}$ is a set of* endogenous

*variables, U is a set of* exogenous *variables, and F is a function from U to V. The arrows* $C_{\mathcal{T}CM}(c, d)$ *are defined through commutative diagrams as illustrated below, where f and f' are the global functions induced by the TCM objects c and d, respectively.*

$$U \xrightarrow{\ h\ } U'$$
$$\downarrow{\scriptstyle f} \qquad\qquad \downarrow{\scriptstyle f'}$$
$$V \xrightarrow{\ g\ } V'$$

*A submodel $c' = \langle U', V', F' \rangle$ of c is any subobject of c. The effect of an intervention on c is given by some submodel c'. Finally, let Y be a variable in V, and let X be a subset of V. The potential outcome in response to an intervention on X modeled by a submodel $c' \hookrightarrow c$ is the solution of Y in the submodel c'.*

A commutative diagram, as the term suggests, is a structure showing the equivalence of two paths. Here, the diagram asserts that $g \circ f = f' \circ h$. In the context of our category $C_{\mathcal{SCM}}$, the arrow $f : U \to V$ is simply an SCM $M$, and $f$ is its induced mapping from exogenous to endogenous variables. Similarly, $f'$ is also the induced function mapping exogenous to endogenous variables for another SCM $M'$. The morphisms $h$ and $g$ are functions on SCMs, which transform one causal model into another. In the specific case we are interested in, these functions define causal interventions, but in general, they may be arbitrary functions.

For completeness, we define a category $C_{\mathcal{SCM}}$ whose objects are indeed SCMs.

**Definition 11.** *The category $C_{\mathcal{SCM}}$ of structural causal models is defined as a collection of objects, each of which is a triple $\langle U, V, F \rangle$ where $V = \{V_1, \dots, V_n\}$ is a set of* endogenous *variables, U is a set of* exogenous *variables, F is a set $\{f_1, \dots, f_n\}$ of "local functions" $f_i : U \cup (V \setminus V_i) \to V_i$ whose composition induces a unique function F from U to V. Let X be a subset of variables in V, and x be a particular realization of X. A submodel $M_x = \langle U, V, F_x \rangle$ of M is the causal model $M_x = \langle U, V, F_x \rangle$, where $F_x = \{f_i : V_i \notin X\} \cup \{X = x\}$. The effect of an action $do(X = x)$ on M is given by the submodel $M_x$. Finally, let Y be a variable in V, and let X be a subset of V. The potential outcome of Y in response to an action $do(X = x)$, denoted $Y_x(u)$, is the solution of Y for the set of equations $F_x$.*

The set of arrows or morphisms between two objects $c$ and $d$ in the category $C_{\mathcal{SCM}}$, denoted $C_{\mathcal{SCM}}(c, d)$, represent ways of transitioning from SCM object $c$ to $d$. For example, if $d$ is a submodel of $c$, then the arrow defines a **do** calculus causal intervention.

## 4  Causal Inference in a Topos Category

We show in this section that a TCM category whose objects are defined as SCMs, and whose arrows correspond to commutative diagrams defining operations on causal models does define a topos. In the next section, we generalize from SCMs to consider more complex causal models over functor categories. Now, we can state the first key result of this paper.

**Theorem 3.** *The category $C_{\mathcal{SCM}}$ forms a topos.*

**Proof:** Since we have previously defined the objects and arrows of the $C_{\mathcal{SCM}}$ category, to show it forms a topos, we need to construct its subobject classifier. First, we need to define what a "subobject" is in the category $C_{\mathcal{SCM}}$. Since SCMs can abstractly be defined as functions, let us assume that the SCM $c$ that defines $f$ is a *submodel* of the SCM $c'$ that induces $g$. We can denote that by defining a commutative diagram as shown below. Let us stress the difference between the commutative diagram shown below Definition 10 for arbitrary functions $g$ and $h$ vs. the one below, where $i$ and $j$ are monic arrows.

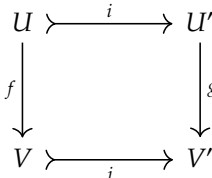

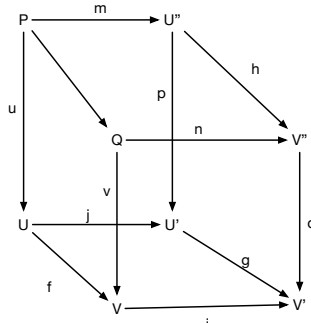 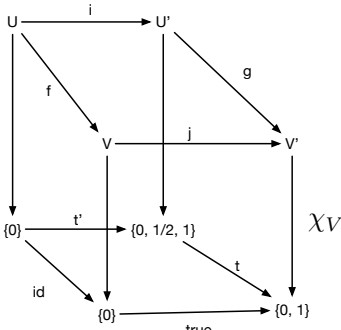

Figure 2: Left: diagram showing that $\mathcal{C}_{SCM}$ has pullbacks. Right: The subobject classifier $\Omega$ for the topos category $\mathcal{C}_{SCM}$ is displayed on the bottom face of this cube.

An element $x \in U'$, which is a particular realization of the exogenous variables in $U'$, can be classified in three ways by defining a characteristic function $\psi$:

1. $x \in U$ – here we set $\psi(x) = \mathbf{1}$.

2. $x \notin U$ but $g(x) \in V$ – here we set $\psi(x) = \frac{1}{2}$.

3. $x \notin U$ and $g(x) \notin V$ – we denote this by $\psi(x) = \mathbf{0}$.

The subobject classifier is illustrated as the bottom face of the cube shown on the right in Figure 2:

- $\mathbf{true}(0) = t'(0) = \mathbf{1}$

- $\mathbf{t} : \{\mathbf{0}, \frac{1}{2}, \mathbf{1}\} \rightarrow \{\mathbf{0}, \mathbf{1}\}$, where $\mathbf{t(0) = 0, t(1) = t(\frac{1}{2}) = 1}$.

- $\chi_V$ is the characteristic function of the exogenous variable set $V$.

- The base of the cube in Figure 2 displays the subobject classifier $\mathbf{T} : \mathbf{1} \rightarrow \mathbf{\Omega}$, where $\mathbf{T} = \langle \mathbf{t'}, \mathbf{true} \rangle$ that maps $\mathbf{1} = \mathbf{id}_{\{0\}}$ to $\Omega = \mathbf{t} : \{0, \frac{1}{2}, 1\} \rightarrow \{0, 1\}$.

This proves that the subobject classifier for the category $\mathcal{C}_{SCM}$ does not have Boolean semantics, but intuitionistic semantics as its subobject classifier $\Omega$ has multiple degrees of "truth", corresponding to the three types of classifications of monic arrows (in regular set theory, there are only two classifications). Moving on to show the other properties of a topos are satisfied, note that the terminal object is simply the identity function $\mathbf{id}_0 : \{0\} \rightarrow \{0\}$. Now, it remains to show that $\mathcal{C}_{SCM}$ has pullbacks and exponential objects.

**Pullbacks in $\mathcal{C}_{SCM}$:** Consider the cube shown on the left in Figure 2. Here, $f$, $g$, and $h$ can be interpreted as three SCMs, each mapping some exogenous variables to some endogenous variables. The arrows $i, j$ ensure that the bottom face of the cube is a commutative diagram, and the arrows $p, q$ ensures the right face of the cube is a commutative diagram. The arrow from $P$ to $Q$ exists because looking at the front face of the cube, $Q$ is the pullback of $i$ and $q$, which must exist because we are in the category of **Sets**, which has all pullbacks. Similarly, the back face of the cube is a pullback of $j$ and $p$, which is again a pullback in **Sets**. Summarizing, $\langle u, v \rangle$ and $\langle m, n \rangle$ are the pullbacks of $\langle i, j \rangle$ and $\langle p, q \rangle$.

**Exponential objects in $\mathcal{C}_{SCM}$:** Now it only remains to check that the category has exponential objects. Let $f : U \rightarrow V$ and $g : U' \rightarrow V'$ be two functions induced by SCM models $M$ and $N$. Then, we need to define the meaning of $g^f$ in $\mathcal{C}_{SCM}$, which we can define as $g^f : X \rightarrow Y$, where $Y = V'^V$, which must exist since **Sets** is a Cartesian closed category that has exponential objects (i.e., $Y$ is simply the set of all functions from $V$ to $V'$). Also, $X$ is the set of all arrows in $\mathcal{C}_{SCM}$ from SCM $M$ to SCM $N$, which is the pair of functions $\langle h, k \rangle$ in the commutative diagram shown below. This finally proves that $\mathcal{C}_{SCM}$ is a topos. □

# 5 Causal Models Over a Topos of Sheaves

We now describe a more general categorical framework for defining causal models as a topos by using the property that Yoneda embeddings of presheaves forms a topos [MacLane and leke Moerdijk, 1994]. To ensure consistent extension into a unique global function, we build on the theory of sheaves [Mac Lane and Moerdijk, 1992], which ensures local functions can be "collated" together to yield a unique global function. In our setting, we will construct sheaves from categories over causal models through the Yoneda embedding $ {\downarrow}(x) : C \to \mathbf{Sets}^{C^{op}} $ and impose a Grothendieck topology.

## 5.1 Grothendieck Topology on Sites

**Definition 12.** *A **sieve** for any object $x$ in any (small) category $C$ is a subobject of its Yoneda embedding ${\downarrow}(x) = C(-, x)$. If $S$ is a sieve on $x$, and $h : y \to x$ is any arrow in category $C$, then*

$$h^*(S) = \{g \mid cod(g) = D, hg \in S\}$$

**Definition 13.** *[Mac Lane and Moerdijk, 1992] A **Grothendieck topology** on a category $C$ is a function $J$ which assigns to each object $x$ of $C$ a collection $J(x)$ of sieves on $x$ such that*

1. *the maximum sieve $t_x = \{f \mid cod(f) = x\}$ is in $J(x)$.*

2. *If $S \in J(x)$ then $h^*(S) \in J(y)$ for any arrow $h : y \to x$.*

3. *If $S \in J(x)$ and $R$ is any sieve on $x$, such that $h^*(R) \in J(y)$ for all $h : y \to x$, then $R \in J(C)$.*

We can now define categories with a given Grothendieck topology as *sites*.

**Definition 14.** *A **site** is defined as a pair $(C, J)$ consisting of a small category $C$ and a Grothendieck topology $J$ on $C$.*

**Definition 15.** *The **subobject classifier** $\Omega$ is defined on any topos $\mathbf{Sets}^{C^{op}}$ as subobjects of the representable functors:*

$$\Omega(x) = \{S \mid S \text{ is a subobject of } C(-, x)\}$$

*and the morphism $\mathbf{true} : 1 \to \Omega$ is $\mathbf{true}(x) = x$ for any representable $x$.*

## 5.2 Universal Property of TCM over Functor Categories

Causal models, like SCMs, must represent both decomposable structure and (probabilistic) semantics. To capture this richer structure, we define TCM over *functor categories*, where every object is a functor that maps structure to semantics. For example, the category of Bayesian networks can be modeled as a functor category [Jacobs et al., 2018, Fritz and Klingler, 2023] from a Markov category to the category **FinStoch** of finite stochastic processes.

**Theorem 4.** *Given a causal functor $A : C \to \mathcal{E}$, such as the Bayesian network functor $F_{CDU}$, from a small category $C$ (e.g., a symmetric monoidal category such as a Markov category) to a cocomplete category $\mathcal{E}$ (e.g., the category **Prob** of probability spaces (see Theorem 6)), the functor $R$ from $\mathcal{E}$ to presheaves, given by (where $c \in C$ and $E \in \mathcal{E}$)*

$$R(E) : c \mapsto \mathbf{Hom}_{\mathcal{E}}(A(c), E)$$

*has a left adjoint $L : \mathbf{Sets}^{C^{op}} \to \mathcal{E}$ defined for each presheaf $P$ in $C^{op}$ as the colimit*

$$L(P) = Colim\left( \int_C P \xrightarrow{\pi_P} C \xrightarrow{A} \mathcal{E} \right)$$

*where $\int_C P$ is the category of elements, whose objects are pairs $(c, p)$, where $c$ is an object of $C$ and $p$ is an element of $P(C)$ (recall $P$ is a presheaf, i.e., a set-valued functor that maps each element $c$ into a set), and its arrows are $(c', p') \to (c, p)$ for any morphism $f : c' \to c$ such that $pc = p'$.*

**Proof:** Essentially, Theorem 4 is stating that there is a pair of adjoint functors $L \vdash R$, defined as:

$$L : \mathbf{Sets}^{C^{op}} \rightleftarrows \mathcal{E} : R$$

As defined earlier, a natural transformation between two functors $\tau : P \to R(E)$ is a family $\{\tau_c\}$ of maps indexed by the objects $c \in C$, where each map $\tau_c$ is defined as the mapping:

$$\tau_c : P(C) \mapsto \mathbf{Hom}_{\mathcal{E}}(A(C), E)$$

which is natural in $c$. $\tau$ can also be defined as a set of arrows of $\mathcal{E}$ as $\{\tau_c(p) : A(c) \to E\}_{(c,p)}$ that is indexed by the objects $(c, p)$ of the category $\int_C P$ of elements of $P$. This fact implies that there is a bijection

$$Nat(P, R(E)) \simeq \mathbf{Hom}_{\mathcal{E}}(LP, E)$$

This bijection being natural in $P$ and in $E$ proves that $L$ is a left adjoint functor to $R$.  □

Now, let us define a general causal functor as mapping from a decomposable symmetric monoidal category (e.g., a Markov category) to the symmetric monoidal category of probability spaces.

**Definition 16.** *A causal functor $F : C \to \mathbf{Prob}$ maps from a general symmetric monoidal category $C$ with a comonoidal "copy-delete" structure (e.g., a CDU category [Jacobs et al., 2018] or a Markov category [Fritz, 2020]) to the category of probability spaces $\mathbf{Prob}$, where each object $(\Omega, \mathcal{F}, \mathbb{P})$ is a probability space, and the arrows are measure-preserving maps, namely $\mathbf{Prob}(c, d)$, where $c = (\Omega_c, \mathcal{F}_c, \mathbb{P}_c)$ and $d = (\Omega_d, \mathcal{F}_d, \mathbb{P}_d)$, where $f \in \mathbf{Prob}(c, d)$ is such that $\mathbb{P}_c(f^{-1}(A)) = \mathbb{P}_d(A)$ for all $A \in \mathcal{F}_d$.*

**Theorem 5.** *For each causal functor $A : C \to \mathcal{E}$ from a small category $C$ defining the structure of a causal model to a cocomplete category $\mathcal{E}$ defining its (probabilistic) semantics, there exists a colimit preserving functor $L : \mathbf{Sets}^{C^{op}} \to \mathcal{E}$ such that $A = L \circ \text{よ}$, where $\text{よ}$ is the Yoneda embedding.*

**Proof**: The proof is just a special case of Corollary 4 on page 43 in [MacLane and leke Moerdijk, 1994]. To emphasize the importance of the co-completeness condition on $\mathcal{E}$, we use the following result from [van Belle, 2024] that the category of probability spaces is co-complete.  □

**Theorem 6.** *[van Belle, 2024] The symmetric monoidal category $\mathbf{Prob}$ has all colimits of non-empty diagrams.*

**Proof:** The proof, given in [van Belle, 2024], shows that $\mathbf{Prob}$ has coproducts and coequalizers. Thus, we can choose $\mathbf{Prob}$ as our cocomplete category $\mathcal{E}$ in Theorem 4.  □

We can finally state the two central results of our paper, the first (Theorem 7) establishes the universal property underlying TCMs, and the second (Theorem 8) shows that causal interventions define a Heyting algebra whose logic is intuitionistic.

**Theorem 7.** *Any causal functor $F : C \to \mathcal{E}$ from a structural causal category $C$ (such as a Markov category) to a semantic cocomplete category $\mathcal{E}$ (such as $\mathbf{Prob}$) factors uniquely through a TCM structure defined by the Yoneda embedding, as given in Theorem 5.*

**Proof:** The proof follows directly from Theorem 4, Theorem 5, Theorem 6, and Definition 16. □

**Definition 17.** *A **Heyting algebra** is a poset with all finite products and coproducts, which is Cartesian closed. That is, a Heyting algebra is a lattice, including bottom and top elements, denoted by $\mathbf{0}$ and $\mathbf{1}$, respectively, which associates to each pair of elements $x$ and $y$ an exponential $y^x$. The exponential is written $x \Rightarrow y$, and defined as an adjoint functor:*

$$z \leq (x \Rightarrow y) \quad \text{if and only if} \quad z \wedge x \leq y$$

In other words, $x \Rightarrow y$ is a least upper bound for all those elements $z$ with $z \wedge x \leq y$. As a concrete example, for a topological space $X$ the set of open sets $O(X)$ is a Heyting algebra. The "law of the excluded middle", meaning $\neg x \vee x = \mathbf{true}$, does not always hold in a Heyting algebra.

**Theorem 8.** *For any TCM category defined as $\hat{C} = \mathbf{Sets}^{C^{op}}$ by the Yoneda embedding $\text{よ}(c)$ of a small causal category $C$, the partially ordered set $Sub_{\hat{C}}(P)$ of subobjects generated by causal interventions on any causal functor defined by the presheaf $P$ is a Heyting algebra.*

**Proof:** This result follows directly from the corresponding result for any category of presheaves (see [MacLane and leke Moerdijk, 1994]), and is based on constructing the complete lattice $Sub(P)$ of all subfunctions of $P$ using a pointwise operation for each object $c \in C$, which can be shown to satisfy an infinite distributive law.  □

## 6 Causal Mitchell-Bénabou Language and its Kripke-Joyal Semantics

The Causal Mitchell-Bénabou language (CMBL) is a typed local set theory [Bell, 1988] whose syntax and semantics is defined using the arrows of the $C_{\mathcal{TCM}}$ topos. The types of CMBL as causal model objects $M$ of $C_{\mathcal{TCM}}$. For each type $M$, we assume the existence of variables $x_M, y_M, \ldots$, where each such variable has as its interpretation the identity arrow $\mathbf{1} : M \to M$. We can construct product objects, such as $A \times B \times C$, where terms like $\sigma$ that define arrows are given the interpretation $\sigma : A \times B \times C \to D$.

- Each variable $x_M$ of type $M$ is a term of type $M$, and its interpretation is the identity $x_M = \mathbf{1} : M \to M$, Here, $M$ may represent an entire SCM, an individual variable such as Overpopulation in Figure 1, or a causal functor mapping a Markov category to the cocomplete **Prob** category.

- Terms $\sigma$ and $\tau$ of types $C$ and $D$ that are interpreted as $\sigma : A \to C$ and $\tau : B \to D$ can be combined to yield a term $\langle \sigma, \tau \rangle$ of type $C \times D$, whose joint interpretation is given as $\langle \sigma p, \tau q \rangle : X \to C \times D$, where $X$ has the required projections $p : X \to A$ and $q : X \to B$. A causal intervention modeled as an arrow $f : X \to Y$ in $C_{TCM}$ can be composed with a term $\sigma : U \to X$ to yield a term of type $Y$ as $f \circ \sigma : U \xrightarrow{\sigma} X \xrightarrow{f} Y$.

- Terms of type $\Omega$ are defined as *formulae* of CMBL and can be combined with the usual logical connectives $\wedge, \vee, \Rightarrow, \neg$ and quantifiers $\forall, \exists$ to obtain further terms again of type $\Omega$. An expression such as $\forall x \, \psi(x, y)$ is interpreted by an arrow $Y \to \Omega$ (since $x$ is not a free variable). A formula $\psi(x, y)$ in the topos $C_{TCM}$ is defined to be *universally valid* in the topos if the corresponding arrow $\psi(x, y) : X \times Y \to \Omega$ factors through **true** $: 1 \to \Omega$. A formula $\psi$ without free variables is interpreted as an arrow $\psi : \mathbf{1} \to \Omega$ and is valid if it coincides with the arrow **true** $: \mathbf{1} \to \Omega$.

- Indirect proofs (i.e., *reductio ad absurdum*) cannot be used in CMBL because the rule of the excluded middle $\psi \vee \neg\psi$ is not in general valid, nor is the axiom of choice generally true. Instead, the rules of intuitionistic predicate calculus need to be used, as illustrated by de Araujo Fernandes and Haeusler [2009].

- The Kripke-Joyal semantics for CMBL is specified using *generalized elements*. We define an element of a causal model by the morphism $x : \mathbf{1} \to M$. Thus, a generalized element $\alpha : N \to M$ represents the "stage of definition" of $M$ by $N$. We specify the semantics of how an TCM model $N$ supports any formula $\phi(\alpha)$, denoted by $N \Vdash \phi(\alpha)$ by $N \Vdash \phi(\alpha)$ if and only if $\text{Im } \alpha \leq \{x | \phi(x)\}$. Stated in the form of a commutative diagram, this "forcing" relationship holds if and only if $\alpha$ factors through $\{x | \phi(x)\}$. See [MacLane and Ieke Moerdijk, 1994, Bell, 1988] for additional details.

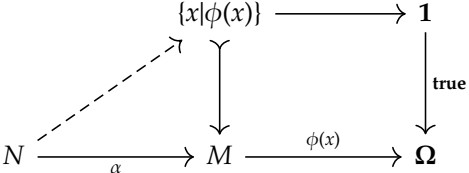

## 7 Limitations and Future Work

There are significant limitations of our TCM framework, which we are currently investigating. We have recently developed an intuitionistic generalization of Pearl's do-calculus termed $j$-stable causal inference, which uses the Lawvere-Tierney topology on a topos defined by a modal operator $j$ on the subobject classifier $\Omega$ [Mahadevan, 2025a]. In this paper, we define an intuitionistic logic called j-do-calculus, where we replace global truth with local truth defined by Kripke-Joyal semantics. We are currently working on another paper [Mahadevan, 2025b] that implements j-do-calculus with well-known causal discovery procedures (e.g., score-based and constraint-based methods) [Zanga and Stella, 2023], and will include experimental results on how to (i) form data-driven $j$-covers (via regime/section constructions), (ii) compute chartwise conditional independences after graph surgeries, and (iii) glue them to certify the premises of the $j$-do rules in practice.

## 8 Acknowledgments

This research has been funded by Adobe Corporation.

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

# Appendix: Category Theory Background

We give an introduction to categories and functors in Section A. Section B gives a brief overview of the theory of sheaves over toposes. Section C defines local set theories, another way to characterize the internal language of a topos. Section E contains an overview of affine CDU and Markov categories, which have been previously studied as categorical models of causality and probability [Jacobs et al., 2018, Fritz and Klingler, 2023].

## A  Categories and Functors

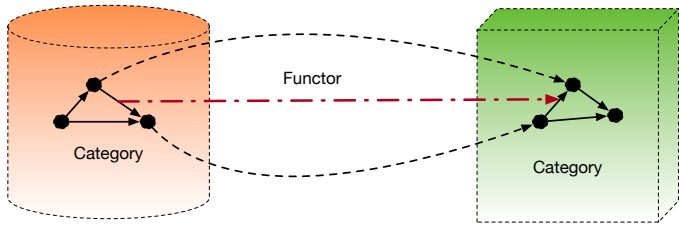

Figure 3: Categories are collections of objects, with a collection of arrows defined between each pair. A functor must map both objects and arrows from a domain category into a co-domain category.

Category theory is perhaps the most transformative rethinking of mathematics since antiquity. Rather than focus on the internals of an object, like the elements of a set, it focuses on the arrows or morphisms that define the interactions between objects. Applied to causal models, it suggests that we can construct languages where individual causal models can be defined as variables in a logical language with intuitionistic semantics. The Yoneda Lemma, one of the most celebrated results in pure mathematics of the 20th century, states that objects are completely defined up to isomorphism purely in terms of their interactions. Applied to SCMs, its implications are nonetheless startling. SCMs in effect can be defined not just in terms of their internal structure, but in terms of how they interact with other SCMs in a category of such objects. Being conditioned to think in terms of internal structure, this statement may seem counterintuitive. But, as category theory has shown in numerous cases, such as in algebraic topology May [1992], we can often understand deep properties of complex objects such as topological spaces by modeling them as combinatorial objects and analyzing their interactions. Our framework builds on the theory of categories, functors, and natural transformations [MacLane, 1971, Riehl, 2017] (see Figure 3 and Figure 4). Many common mathematical structures – groups, rings, modules, measurable spaces, topological spaces etc. – form categories. More interesting categories for AI are those associated with compositional machine learning models, such as deep learning [Fong et al., 2019], where a symmetric monoidal category `Learn` is defined that combines an `Implement` routine that maps an input $A$ into some output $B$ parameterized by some parameter $P$, and an `Update` routine that given an input-output pair $A, B$ and a parameter $P$, returns a new parameter $P$. Causal models, such as DAGs, structural equation models etc. can be straightforwardly mapped into categories.

More formally, a category $C$ is a collection of abstract *objects* $c \in C$. Anything technically can count as an object, from a variable in a causal model to an entire model itself. Each category $C$ is additionally specified by a set of arrows or morphisms $C(c, d)$ between each pair of objects $c$ and $d$. There is an identity arrow $1_c \in C(c, c)$. Arrows compose in the obvious way, inducing a function $C(c, d) \times C(d, e) \to C(c, e)$. An *initial object* $c$ in category $C$ defined as one inducing a unique arrow from $c$ to every object in category $C$. A *terminal* object, usually denoted by $\mathbf{1}$, is one that defines a unique arrow from every object $c$ in category $C$ into $\mathbf{1}$. An object $c$ is isomorphic to another object $d$, denoted $c \simeq d$, if two arrows $f : c \to d$ and $g : d \to c$ exist, such that $g \circ f = 1_c$, and $f \circ g = 1_d$. A *functor* $F : C \to \mathcal{D}$ between two categories $C$ and $\mathcal{D}$ is specified by an *object function* mapping each $c \in C$ to $Fc \in \mathcal{D}$, and an *arrows function* mapping each arrow $f \in C(c, d)$ to $Ff \in \mathcal{D}(Fc, Fd)$. Functors come in two varieties – covariant and contravariant – the latter acts on the domain category by reversing the arrows. Given

| Set theory | Topos theory |
|---|---|
| set | object |
| subset | subobject |
| truth values $\{0, 1\}$ | subobject classifier $\Omega$ |
| power set $P(A) = 2^A$ | power object $P(A) = \Omega^A$ |
| bijection | isomorphims |
| injection | monic arrow |
| surjection | epic arrow |
| singleton set $\{*\}$ | terminal object $\mathbf{1}$ |
| empty set $\emptyset$ | initial object $\mathbf{0}$ |
| elements of a set $X$ | morphism $f : \mathbf{1} \to X$ |
| - | functors, natural transformations |
| - | limits, colimits, adjunctions |

Figure 4: A topos is a category that generalizes set theory: subsets become subobjects and the characteristic function for a subset, which is Boolean, turns into a subobject classifier that may have multiple "degrees of truth".

any two functors $F : C \to D$ and $G : C \to D$ between the same pair of categories, we can define a mapping between $F$ and $G$ that is referred to as a natural transformation. These are defined through a collection of mappings, one for each object $c$ of $C$, thereby defining a morphism in $D$ for each object in $C$.

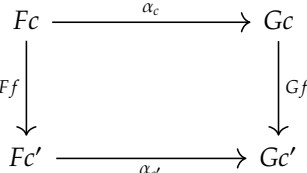

The celebrated Yoneda Lemma [MacLane, 1971] states that any set-valued functor $F : C \to \mathbf{Sets}$ can be modeled in terms of natural transformations between $F$ and the Yoneda embedding $C(x, -)$ (or $C(-, x)$) of a *representable* functor. In essence the action of a set valued functor $F$ on an object $x$ is completely determined by its natural transformations with $C(-, x)$. What are natural transformations? These specify how functors interact, much like how objects interact through arrows. A remarkable property of category theory is that the set of all sets is not a set, but the category of all categories is indeed a category, where the arrows are functors, and the objects are categories.

**Definition 18.** *Given categories $C$ and $D$, and functors $F, G : C \to D$, a **natural transformation** $\alpha : F \Rightarrow G$ is defined by the following data:*

- *an arrow $\alpha_c : Fc \to Gc$ in $D$ for each object $c \in C$, which together define the components of the natural transformation.*

- *For each morphism $f : c \to c'$, the following commutative diagram holds true:*

$$
\begin{array}{ccc}
Fc & \xrightarrow{\alpha_c} & Gc \\
{\scriptstyle Ff}\downarrow & & \downarrow{\scriptstyle Gf} \\
Fc' & \xrightarrow{\alpha_{c'}} & Gc'
\end{array}
$$

*A **natural isomorphism** is a natural transformation $\alpha : F \Rightarrow G$ in which every component $\alpha_c$ is an isomorphism.*

## A.1 Yoneda Lemma

The Yoneda Lemma states that the set of all morphisms into an object $d$ in a category $C$, denoted as $\mathbf{Hom}_C(-, d)$ and called the *contravariant functor* (or presheaf), is sufficient to define $d$ up to isomorphism. The category of all presheaves forms a *category of functors*, and is denoted $\hat{C} = \mathbf{Set}^{C^{op}}$. The Yoneda lemma plays a crucial role in this paper because it defines the concept of a *universal representation* in category theory. We first show that associated with universal arrows is the corresponding induced isomorphisms between $\mathbf{Hom}$ sets of morphisms in categories. This universal property then leads to the Yoneda lemma.

$$
\begin{array}{ccc}
D(r,r) & \xrightarrow{\ \phi_r\ } & C(c, Sr) \\
\downarrow{\scriptstyle D(r,f')} & & \downarrow{\scriptstyle C(c, Sf')} \\
D(r,d) & \xrightarrow{\ \phi_d\ } & C(c, Sd)
\end{array}
$$

As the two paths shown here must be equal in a commutative diagram, we get the property that a bijection between the $\mathbf{Hom}$ sets holds precisely when $\langle r, u : c \to Sr \rangle$ is a universal arrow from $c$ to $S$. Note that for the case when the categories $C$ and $D$ are small, meaning their $\mathbf{Hom}$ collection of arrows forms a set, the induced functor $\mathbf{Hom}_C(c, S-)$ to $\mathbf{Set}$ is isomorphic to the functor $\mathbf{Hom}_D(r, -)$. This type of isomorphism defines a universal representation, and is at the heart of the causal reproducing property (CRP) defined below.

**Lemma 2. Yoneda lemma**: *For any functor $F : C \to \mathbf{Set}$, whose domain category $C$ is "locally small" (meaning that the collection of morphisms between each pair of objects forms a set), any object $c$ in $C$, there is a bijection*

$$
Hom(C(c, -), F) \simeq Fc
$$

*that defines a natural transformation $\alpha : C(c, -) \Rightarrow F$ to the element $\alpha_c(1_c) \in Fc$. This correspondence is natural in both $c$ and $F$.*

There is of course a dual form of the Yoneda Lemma in terms of the contravariant functor $C(-, c)$ as well using the natural transformation $C(-, c) \Rightarrow F$. A very useful way to interpret the Yoneda Lemma is through the notion of universal representability through a covariant or contravariant functor.

**Definition 19.** *A **universal representation** of an object $c \in C$ in a category $C$ is defined as a contravariant functor $F$ together with a functorial representation $C(-, c) \simeq F$ or by a covariant functor $F$ together with a representation $C(c, -) \simeq F$. The collection of morphisms $C(-, c)$ into an object $c$ is called the **presheaf**, and from the Yoneda Lemma, forms a universal representation of the object.*

## A.2 Universal Properties in Categories

A fundamental principle of category theory is to characterize objects by universal properties. Take the Cartesian product of two sets. The conventional way to define this product of two sets is as the set of ordered pairs, one drawn from each set. But this definition does not specify its universal property. We explain how to define universal properties below, which will be essential to understanding a topos. A key distinguishing feature of category theory is the use of diagrammatic reasoning. However, diagrams are also viewed more abstractly as functors mapping from some indexing category to the actual category. Diagrams are useful in understanding universal constructions, such as limits and colimits of diagrams. To make this somewhat abstract definition concrete, let us look at some simpler examples of universal properties, including co-products and quotients (which in set theory correspond to disjoint unions). Coproducts refer to the universal property of abstracting a group of elements into a larger one.

Before we formally the concept of limit and colimits, we consider some examples. These notions generalize the more familiar notions of Cartesian products and disjoint unions in the category of **Sets**, the notion of meets and joins in the category **Preord** of preorders, as well

as the least upper bounds and greatest lower bounds in lattices, and many other concrete examples from mathematics.

**Example 1.** *If we consider a small "discrete" category $\mathcal{D}$ whose only morphisms are identity arrows, then the colimit of a functor $\mathcal{F} : \mathcal{D} \to C$ is the* categorical coproduct *of $\mathcal{F}(D)$ for D, an object of category D, is denoted as*

$$Colimit_{\mathcal{D}}F = \bigsqcup_{D} \mathcal{F}(D)$$

*In the special case when the category C is the category* **Sets***, then the colimit of this functor is simply the disjoint union of all the sets F(D) that are mapped from objects $D \in \mathcal{D}$.*

**Example 2.** *Dual to the notion of colimit of a functor is the notion of* limit*. Once again, if we consider a small "discrete" category $\mathcal{D}$ whose only morphisms are identity arrows, then the limit of a functor $\mathcal{F} : \mathcal{D} \to C$ is the* categorical product *of $\mathcal{F}(D)$ for D, an object of category D, is denoted as*

$$limit_{\mathcal{D}}F = \prod_{D} \mathcal{F}(D)$$

*In the special case when the category C is the category* **Sets***, then the limit of this functor is simply the Cartesian product of all the sets F(D) that are mapped from objects $D \in \mathcal{D}$.*

Category theory relies extensively on *universal constructions*, which satisfy a universal property. One of the central building blocks is the identification of universal properties through formal diagrams. Before introducing these definitions in their most abstract form, it greatly helps to see some simple examples. We can illustrate the limits and colimits in diagrams using pullback and pushforward mappings.

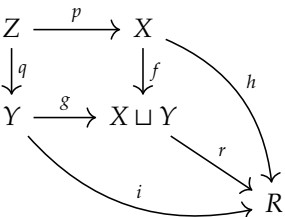

An example of a universal construction is given by the above commutative diagram, where the coproduct object $X \sqcup Y$ uniquely factorizes any two mappings $h : X \to R$ and $i : Y \to R$, such that any mapping $i : Y \to R$, so that $h = r \circ f$, and furthermore $i = r \circ g$. Co-products are themselves special cases of the more general notion of co-limits. Figure 5 illustrates the fundamental property of a *pullback*, which along with *pushforward*, is one of the core ideas in category theory. The pullback square with the objects $U, X, Y$ and $Z$ implies that the composite mappings $g \circ f'$ must equal $g' \circ f$. In this example, the morphisms $f$ and $g$ represent a *pullback* pair, as they share a common co-domain Z. The pair of morphisms $f', g'$ emanating from $U$ define a *cone*, because the pullback square "commutes" appropriately. Thus, the pullback of the pair of morphisms $f, g$ with the common co-domain Z is the pair of morphisms $f', g'$ with common domain $U$. Furthermore, to satisfy the universal property, given another pair of morphisms $x, y$ with common domain $T$, there must exist another morphism $k : T \to U$ that "factorizes" $x, y$ appropriately, so that the composite morphisms $f' k = y$ and $g' k = x$. Here, $T$ and $U$ are referred to as *cones*, where $U$ is the limit of the set of all cones "above" Z. If we reverse arrow directions appropriately, we get the corresponding notion of pushforward. So, in this example, the pair of morphisms $f', g'$ that share a common domain represent a pushforward pair. For any set-valued functor $\delta : S \to$ **Sets**, the Grothendieck category of elements $\int \delta$ can be shown to be a pullback in the diagram of categories. Here, **Set**$_*$ is the category of pointed sets, and $\pi$ is a projection that sends a pointed set $(X, x \in X)$ to its underlying set X.

We can now proceed to define limits and colimits more generally. We define a *diagram F of shape J* in a category C formally as a functor $F : J \to C$. We want to define the somewhat abstract concepts of *limits* and *colimits*, which will play a central role in this paper in defining

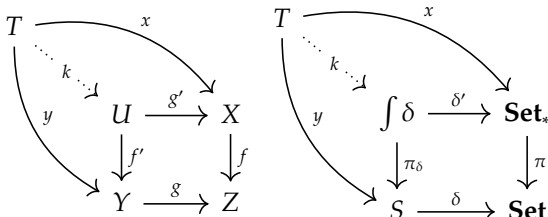

Figure 5: (**Left**) Universal Property of pullback mappings. (**Right**) The Grothendieck category of elements $\int \delta$ of any set-valued functor $\delta : S \to$ **Set** can be described as a pullback in the diagram of categories. Here, **Set**$_*$ is the category of pointed sets $(X, x \in X)$, and $\pi$ is the "forgetful" functor that sends a pointed set $(X, x \in X)$ into the underlying set $X$.

a topos. A convenient way to introduce these concepts is through the use of *universal cones* that are *over* and *under* a diagram.

For any object $c \in C$ and any category $J$, the *constant functor* $c : J \to C$ maps every object $j$ of $J$ to $c$ and every morphism $f$ in $J$ to the identity morphisms $1_c$. We can define a constant functor embedding as the collection of constant functors $\Delta : C \to C^J$ that send each object $c$ in $C$ to the constant functor at $c$ and each morphism $f : c \to c'$ to the constant natural transformation, that is, the natural transformation whose every component is defined to be the morphism $f$.

**Definition 20.** *A* **cone over** *a diagram* $F : J \to C$ *with the* **summit** *or* **apex** $c \in C$ *is a natural transformation* $\lambda : c \Rightarrow F$ *whose domain is the constant functor at* $c$. *The components* $(\lambda_j : c \to Fj)_{j \in J}$ *of the natural transformation can be viewed as its* **legs**. *Dually, a* **cone under** *F with* **nadir** $c$ *is a natural transformation* $\lambda : F \Rightarrow c$ *whose legs are the components* $(\lambda_j : Fj \to c)_{j \in J}$.

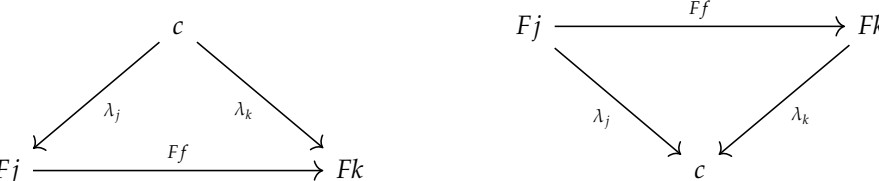

Cones under a diagram are referred to usually as *cocones*. Using the concept of cones and cocones, we can now formally define the concept of limits and colimits more precisely.

**Definition 21.** *For any diagram* $F : J \to C$, *there is a functor*

$$Cone(-, F) : C^{op} \to \textbf{Set}$$

*which sends* $c \in C$ *to the set of cones over F with apex c. Using the Yoneda Lemma, a* **limit** *of F is defined as an object* $\lim F \in C$ *together with a natural transformation* $\lambda : \lim F \to F$, *which can be called the* **universal cone** *defining the natural isomorphism*

$$C(-, \lim F) \simeq Cone(-, F)$$

*Dually, for colimits, we can define a functor*

$$Cone(F, -) : C \to \textbf{Set}$$

*that maps object* $c \in C$ *to the set of cones under F with nadir c. A* **colimit** *of F is a representation for Cone(F, −). Once again, using the Yoneda Lemma, a colimit is defined by an object* $ColimF \in C$ *together with a natural transformation* $\lambda : F \to colimF$, *which defines the* **colimit cone** *as the natural isomorphism*

$$C(colimF, -) \simeq Cone(F, -)$$

Limit and colimits of diagrams over arbitrary categories can often be reduced to the case of their corresponding diagram properties over sets. One important stepping stone is to understand how functors interact with limits and colimits.

**Definition 22.** *For any class of diagrams $K : J \to C$, a functor $F : C \to D$*

- **preserves** *limits if for any diagram $K : J \to C$ and limit cone over $K$, the image of the cone defines a limit cone over the composite diagram $FK : J \to D$.*

- **reflects** *limits if for any cone over a diagram $K : J \to C$ whose image upon applying $F$ is a limit cone for the diagram $FK : J \to D$ is a limit cone over $K$*

- **creates** *limits if whenever $FK : J \to D$ has a limit in $D$, there is some limit cone over $FK$ that can be lifted to a limit cone over $K$ and moreoever $F$ reflects the limits in the class of diagrams.*

To interpret these abstract definitions, it helps to concretize them in terms of a specific universal construction, like the pullback defined above $c' \to c \leftarrow c''$ in $C$. Specifically, for pullbacks:

- A functor $F$ **preserves pullbacks** if whenever $p$ is the pullback of $c' \to c \leftarrow c''$ in $C$, it follows that $Fp$ is the pullback of $Fc' \to Fc \leftarrow Fc''$ in $D$.
- A functor $F$ **reflects pullbacks** if $p$ is the pullback of $c' \to c \leftarrow c''$ in $C$ whenever $Fp$ is the pullback of $Fc' \to Fc \leftarrow Fc''$ in $D$.
- A functor $F$ **creates pullbacks** if there exists some $p$ that is the pullback of $c' \to c \leftarrow c''$ in $C$ whenever there exists a $d$ such that $d$ is the pullback of $Fc' \to Fc \leftarrow Fc''$ in $F$.

### A.3 Symmetric Monoidal Categories

Categorical models of causality [Fong, 2012, Fritz and Klingler, 2023, Jacobs et al., 2018, Mahadevan, 2023] are usually defined over symmetric monoidal categories, which we briefly review now [MacLane, 1971, Richter, 2020].

**Definition 23.** *A* **monoidal category** *is a category $C$ together with a functor $\otimes : C \times C \to C$, an identity object $e$ of $C$ and natural isomorphisms $\alpha, \lambda, \rho$ defined as:*

$$
\begin{aligned}
\alpha_{C_1, C_2, C_3} : C_1 \otimes (C_2 \otimes C_3) &\cong (C_1 \otimes C_2) \otimes C_2, \\
\lambda_C : e \otimes C &\cong C, \\
\rho : C \otimes e &\cong C,
\end{aligned}
$$

The natural isomorphisms must satisfy coherence conditions called the "pentagon" and "triangle" diagrams [MacLane, 1971]. An important result shown in [MacLane, 1971] is that these coherence conditions guarantee that all well-formed diagrams must commute. There are many natural examples of monoidal categories, the simplest one being the category of finite sets, termed **FinSet** in [Fritz, 2020], where each object $C$ is a set, and the tensor product $\otimes$ is the Cartesian product of sets, with functions acting as arrows. Deterministic causal models can be formulated in the category **FinSet**. Other examples include the category of sets with relations as morphisms, and the category of Hilbert spaces [Heunen and Vicary, 2019]. Markov categories [Fritz and Klingler, 2023] are monoidal categories, where the identity element $e$ is also a terminal object, meaning there is a unique "delete" morphism $d_e : X \to e$ associated with each object $X$. [Fritz, 2020] shows they form a unifying foundation for probabilistic and statistical reasoning.

**Definition 24.** *A* **symmetric monoidal category** *is a monoidal category $(C, \otimes, e, \alpha, \lambda, \rho)$ together with a natural isomorphism*

$$\tau_{C_1, C_2} : C_1 \otimes C_2 \cong C_2 \otimes C_1, \text{ for all objects } C_1, C_2$$

*where $\tau$ satisfies the additional conditions: for all objects $C_1, C_2$ $\tau_{C_2, C_1} \circ \tau_{C_1, C_2} \cong 1_{C_1 \otimes C_2}$, and for all objects $C$, $\rho_C = \lambda_C \circ \tau_{C,e} : C \otimes e \cong C$.*

An additional hexagon axiom is required to ensure that the $\tau$ natural isomorphism is compatible with $\alpha$. The $\tau$ operator is called a "swap" in Markov categories [Fritz, 2020]. In most cases of interest in AI, the symmetric monoidal categories are *enriched* over some convenient base category $\mathcal{V}$, including vector spaces, or preorders such as the unit interval $[0, 1]$, where the unique morphism from $a \rightarrow b$ exists if and only if $a \leq b$. 5

**Definition 25.** *A **V-enriched category** consists of a regular category $C$, such that for each pair of objects $x$ and $y$ in $C$, the morphisms $C(x, y) \in \mathcal{V}$, often referred to as a $\mathcal{V}$-hom object. For the case when $(calV, \leq, \otimes, 1)$ is a commutative monoidal preorder, we have the following conditions*

- $1 \leq C(x, x)$

- $C(y, z) \otimes C(x, y) \leq C(x, z)$

## A.4 Topos Category

To help build intuition for a topos, let's understand why sets are special as a category. They have all limits and colimits, meaning that one can always construct the categorical product of two sets as the Cartesian product, and the disjoint union as the coproduct. They have all pullbacks, which are commutative diagrams that define in essence a categorical definition of products and coproducts. In terms of the Yoneda Lemma, a universal property is either an initial or final property in a category of diagrams (functors). The product is the final object in a category of diagrams, and the coproduct is the initial object. Every concept in category theory essentially reduces down to these simple notions. Sets also have exponential objects: the set of all functions between two sets is once again a set! They also have a subobject classifier: each set has subsets, defining its parts, which correspond to a characteristic function that evaluates to true for elements in the subset. Topos theory generalizes all these properties to categories. It is important to causal inference because a causal intervention defines a subobject of an arbitrary object. A causal intervention on an SCM produces a subobject of that SCM object.

To define a topos, we need to go through a few more definitions.

**Definition 26.** *An object $x$ in a category $C$ is called **initial** if there is a unique morphism $f : x \rightarrow y$ to every other object $y$ in the category. Dually, an object is called **final** if there is a unique morphism $f : y \rightarrow x$ into $x$.*

In the category **Sets**, the null or empty set $\emptyset$ is the initial and the single element set $\{*\}$ is the final object. From an empty set, there is only one function possible to any other set, namely the empty function. From any set there is exactly one function into the single element set. A topos generalizes the property of subobject classifiers in **Sets**. Given any subset $S \subset X$, we can define $S$ as the monic arrow $S \hookrightarrow X$ defined by the inclusion of $S$ in $X$, or as the characteristic function $\phi_S$ that is equal to 1 for all elements $x \in X$ that belong to $S$, and takes the value 0 otherwise. We can define the set $\mathbf{2} = \{0, 1\}$ and treat **true** as the inclusion $\{1\}$ in $\mathbf{2}$. The characteristic function $\phi_S$ can then be defined as the pullback of **true** along $\phi_S$.

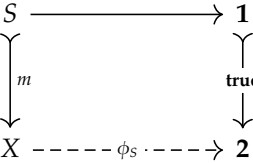

We can now define subobject classifiers in a category $C$ as follows.

**Definition 27.** *In a category $C$ with finite limits, a **subobject classifier** is a monic arrow **true** : $\mathbf{1} \rightarrow \Omega$, such that to every other monic arrow $S \hookrightarrow X$ in $C$, there is a unique arrow $\phi$ that forms the following pullback square:*

$$S \longrightarrow \mathbf{1}$$

with vertical arrows $m$ and **true**, and bottom $X \dashrightarrow^{\phi} \Omega$.

This definition can be rephrased as saying that the subobject functor is representable. In other words, a subobject of an object $x$ in a category $C$ is an equivalence class of monic arrows $m : S \hookrightarrow x$.

**Definition 28.** *A* **topos** *is a category $\mathcal{E}$ with*

1. *A pullback for every diagram $X \to B \leftarrow Y$.*

2. *A terminal object $\mathbf{1}$.*

3. *An object $\Omega$ and a monic arrow* **true** $: 1 \to \Omega$ *such that any monic $m : S \hookrightarrow B$, there is a unique arrow $\phi : B \to \Omega$ in $\mathcal{E}$ for which the following square is a pullback:*

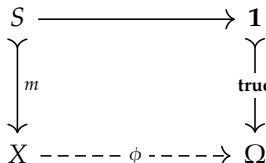

4. *To each object $x$ an object $Px$ and an arrow $\epsilon_x : x \times Px \to \Omega$ such that for every arrow $f : x \times y \to \Omega$, there is a unique arrow $g : y \to Px$ for which the following diagrams commute:*

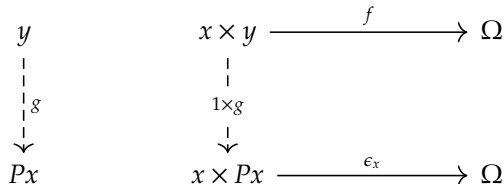

Let us understand these definitions in the category of **Sets**. Clearly, the single point set $\{\bullet\}$ is a terminal object for **Sets**, because there is a unique function from any set $S$ to a single element set $\bullet$, and the categorical product of two sets $A \times B$ is just the Cartesian product. Furthermore, given two sets $A$ and $B$, we can define $B^A$ as the exponential object representing the set of all functions $f : A \to B$. We can define exponential objects in any category more generally as follows.

**Definition 29.** *Given any category $C$ with products, for a fixed object $x$ in $C$, we can define the functor*

$$x \times - : \to C$$

*If this functor has a right adjoint, which can be denoted as*

$$(-)^x : C \to C$$

*then we say $x$ is an* **exponentiable** *object of $C$.*

**Definition 30.** *A category $C$ is* **Cartesian closed** *if it has finite products (which is equivalent to saying it has a terminal object and binary products) and if all objects in $C$ are* exponentiable.

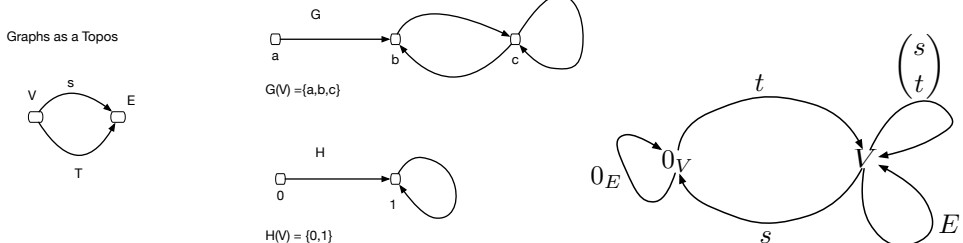

Figure 6: Causal models over arbitrary directed graphs define a topos. The subobject classifier is illustrated on the right.

## A.5 The Topos of Causal Graphs

To begin with a relatively elementary construction involving sheaves, Figure 6 illustrates how directed graphs, widely used in causal modeling, can be modeled as a topos that only has two objects: a generic "vertex" object $V$, representing an abstract causal variable, and a generic "edge" object $E$, denoting an abstract causal path. Any actual graph, such as the two shown in the figure, are covariant functors from $C_\Gamma$, the graph topos, to the actual graphs (we can treat graphs equivalently as contravariant functors by reversing the edges from $V$ to $E$). Sample object mappings of $V$ for the two graphs are shown. For the topos category $C_\Gamma$, the "representable functor" is defined as the presheaf $C_\Gamma(-, c)$ for each object $c$ in the category $\Gamma$, which means the set of all morphisms going *into* object $c$. Let us calculate the representable functors for the topos of graphs $C_\Gamma$. Since $C_\Gamma$ has only two objects, $V$ and $E$, the representable functors are given as the sets:

$$C_\Gamma(-, V), \quad C_\Gamma(-, E)$$

Since $C_\Gamma$ has no arrows from object $E$ to object $V$, we can easily check that the representable functor $C_\Gamma(-, V)$ is given as:

$$C_\Gamma(V, V) = \{1_V\}, \quad C_\Gamma(E, V) = \emptyset$$

where $1_V$ is the self-loop arrow that maps object $V$ to itself. On the other hand, the representable functor $C_\Gamma(-, E)$ is defined as:

$$C_\Gamma(V, E) = \{s, t\}, \quad C_\Gamma(E, E) = \{1_E\}$$

Additional details, like constructing exponential objects $H^G$ for the $C_\Gamma$ category, are given in [Vigna, 2003]. For the subobject classifier, the idea is as follows. For any causal model $N$ represented as a submodel of a more complex causal model $M$, defined by a monic arrow $m : N \hookrightarrow M$, the generalization of the usual set-theoretic characteristic function is the classifying map $\chi_m : N \to \Omega$. As in Theorem 3, $\Omega$ is not Boolean, but has multiple "degrees of truth" (see Figure 6):

1. Causal variables not in $N$ are mapped to $O_V$.
2. Causal variables in $N$ are mapped to $V$.
3. If an edge is not in $N$, four cases emerge:
   (a) An arc whose source and target are not in $N$ is mapped to $0_E$.
   (b) An arc whose source is in $N$, but target is not, is mapped to an edge $s$.
   (c) An arc whose target is in $N$, but source is not, is mapped to an edge $t$.
   (d) An arc having both source and target in $N$ is mapped to $\binom{s}{t}$.[2]

# B   Sheaves and Toposes: Categories of Functors

A general way to construct a topos category is through covariant Yoneda embeddings $よ : C \to \mathbf{Set}^C$, or contravariant Yoneda embeddings $よ : C \to \mathbf{Set}^{C^{op}}$. In simpler terms,

---

[2] $\binom{s}{t} : V + V \to E$ is a map defined by the universal property of coproducts.

each object $x$ in the category $C$ is either mapped to the functor $C(x, -) : C \to$ **Sets** or $C(-, x) : C^{op} \to$ **Sets**. These structures are called presheaves or copresheaves. To construct a proper sheaf, we need to include an additional condition that is illustrated in Figure 7. The sheaf condition plays an important role in many applications of machine learning, from dimensionality reduction [McInnes et al., 2018] to causal inference [Mahadevan, 2023]. Mac Lane and Moerdijk [1992] provides an excellent overview of sheaves and topoi, and how remarkably they unify much of mathematics, from geometry to logic and topology. For causal inference, the structure of causality shapes the arrows in a causal model, such as a Markov category, and that imposes a Grothendieck topology with an associated internal logic. Logic and causality are interwoven in ths sense.

Figure 7: Two applications of sheaf theory in AI: (top) minimizing travel costs in weighted graphs satisfies the sheaf principle, one example of which is the Bellman optimality principle in reinforcement learning [Bertsekas, 2019] (bottom): Approximating a function over a topological space must satisfy the sheaf condition.

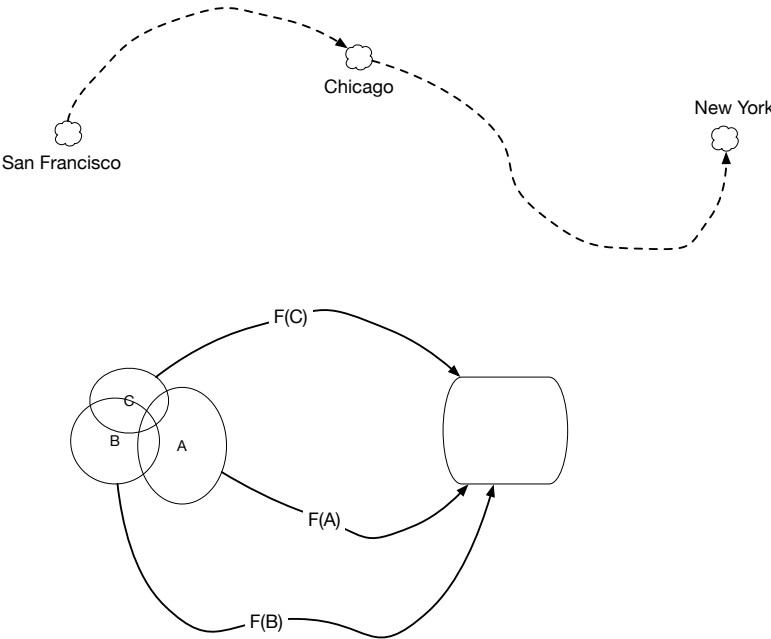

Figure 7 gives two concrete examples of sheaves (in both cases, these are enriched sheaves). In a minimum cost transportation problem, say reinforcement learning [Bertsekas, 2019], any optimal solution has the property that any restriction of the solution must also be optimal. In RL, this sheaf principle is codified by the Bellman equation, and leads to the fundamental principle of dynamic programming [Bertsekas, 2019]. Consider routing candy bars from San Francisco to New York city. If the cheapest way to route candy bars is through Chicago, then the restriction of the overall route to the (sub) route from Chicago to New York City must also be optimal, otherwise it is possible to find a shortest overall route by switching to a lower cost route. Similarly, in function approximation with real-valued functions $F : C \to \mathbb{R}$, where $C$ is the category of topological spaces, the (sub)functions $F(A)$, $F(B)$ and $F(C)$ restricted to the open sets $A$, $B$ and $C$ must agree on the values they map the elements in the intersections $A \cap B$, $A \cap C$, $A \cap B \cap C$ and so on.

Sheaves can be defined over arbitrary categories, and we introduce the main idea by focusing on the category of sheaves over **Sets**.

**Definition 31.** *[Mac Lane and Moerdijk, 1992] A* **sheaf** *of sets F on a topological space X is a functor $F : O^{op} \to$ **Sets** such that each open covering $U = \bigcup_i U_i, i \in I$ of an open set O of X yields an equalizer diagram*

Figure 8: Sieves are subobjects of of $ʎ(x)$ Yoneda embeddings of a category $C$, which generalizes the concept of sheaves over sets in Figure 7.

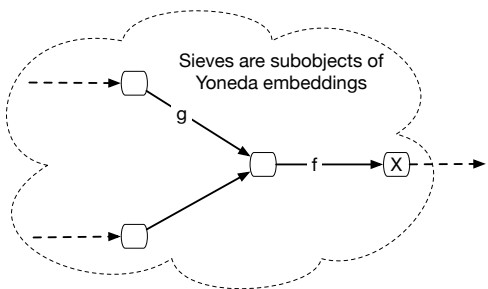

$$FU \xrightarrow{e} \prod_i FU_i \underset{q}{\overset{p}{\rightrightarrows}} \prod_{i,j} F(U_i \cap U_j)$$

The above definition succinctly generalizes the idea in SCMs of combining local functions to get a unique global function.

**Definition 32.** *The category $Sh(X)$ of sheaves over a space $X$ is a full subcategory of the functor category* **Sets**$^{O(X)^{op}}$.

## B.1 Grothendieck Topologies

We can generalize the notion of sheaves to arbitrary categories using the Yoneda embedding $ʎ(x) = C(-, x)$. We explain this generalization in the context of a more abstract topology on categories called the *Grothendieck topology* defined by *sieves*. A sieve can be viewed as a *subobject $S \subseteq ʎ(x)$ in the presheaf* **Sets**$^{C^{op}}$, but we can define it more elegantly as a family of morphisms in $C$, all with codomain $x$ such that

$$f \in S \implies f \circ g \in S$$

Figure 8 illustrates the idea of sieves. A simple way to think of a sieve is as a *right ideal*. We can define that more formally as follows:

**Definition 33.** *If $S$ is a sieve on $x$, and $h : D \to x$ is any arrow in category $C$, then*

$$h^*(S) = \{g \mid cod(g) = D, hg \in S\}$$

**Definition 34.** *[Mac Lane and Moerdijk, 1992] A* **Grothendieck topology** *on a category $C$ is a function $J$ which assigns to each object $x$ of $C$ a collection $J(x)$ of sieves on $x$ such that*

1. *the maximum sieve $t_x = \{f|cod(f) = x\}$ is in $J(x)$.*

2. *If $S \in J(x)$ then $h^*(S) \in J(D)$ for any arrow $h : D \to x$.*

3. *If $S \in J(x)$ and $R$ is any sieve on $x$, such that $h^*(R) \in J(D)$ for all $h : D \to x$, then $R \in J(C)$.*

We can now define categories with a given Grothendieck topology as *sites*.

**Definition 35.** *A* **site** *is defined as a pair $(C, J)$ consisting of a small category $C$ and a Grothendieck topology $J$ on $C$.*

An intuitive way to interpret a site is as a generalization of the notion of a topology on a space $X$, which is defined as a set $X$ together with a collection of open sets $O(X)$. The sieves on a category play the role of "open sets".

## B.2 Mitchell-Bénabou Language

We define the Mitchell-Bénabou language (MBL), a typed local set theory (see Section C) associated with a causal topos. Given the topos category $C_\Omega$, we define the types of MBL as causal model objects $M$ of $C_\Omega$. For each type $M$ (e.g., an SCM), we assume the existence of variables $x_M, y_M, \ldots$, where each such variable has as its interpretation the identity arrow $\mathbf{1} : M \to M$. We can construct product objects, such as $A \times B \times C$, where terms like $\sigma$ that define arrows are given the interpretation $\sigma : A \times B \times C \to D$. We can inductively define the terms and their interpretations in a topos category as follows (see [Mac Lane and Moerdijk, 1992] for additional details):

- Each variable $x_M$ of type $M$ is a term of type $M$, and its interpretation is the identity $x_M = \mathbf{1} : M \to M$ (e.g., $M$ may be an SCM or a causal model on a Markov category).

- Terms $\sigma$ and $\tau$ of types $C$ and $D$ that are interpreted as $\sigma : A \to C$ and $\tau : B \to D$ can be combined to yield a term $\langle \sigma, \tau \rangle$ of type $C \times D$, whose joint interpretation is given as

$$\langle \sigma p, \tau q \rangle : X \to C \times D$$

  where $X$ has the required projections $p : X \to A$ and $q : X \to B$.

- Terms $\sigma : A \to B$ and $\tau : C \to B$ of the same type $B$ yield a term $\sigma = \tau$ of type $\Omega$, interpreted as

$$(\sigma = \tau) : W \xrightarrow{\langle \sigma p, \tau q \rangle} B \times B \xrightarrow{\delta_B} \Omega$$

  where $\delta_B$ is the characteristic map of the diagonal functor $\Delta B \to B \times B$. These diagonal maps correspond to the "copy" procedure in Markov categories [Fritz, 2020].

- Arrows $f : A \to B$ and a term $\sigma : C \to A$ of type $A$ can be combined to yield a term $f \circ \sigma$ of type $B$, whose interpretation is naturally a composite arrow:

$$f \circ \sigma : C \xrightarrow{\sigma} A \xrightarrow{f} B$$

- For exponential objects, terms $\theta : A \to B^C$ and $\sigma : D \to C$ of types $B^C$ and $C$, respectively, combine to give an "evaluation" map of type $B$, defined as

$$\theta(\sigma) : W \to B^C \times C \xrightarrow{e} B$$

  where $e$ is the evaluation map, and $W$ defines a map $\langle \theta p, \sigma q \rangle$, where once again $p : W \to A$ and $q : W \to D$ are projection maps.

- Terms $\sigma : A \to B$ and $\tau : D \to \Omega^B$ combine to yield a term $\sigma \in \tau$ of type $\Omega$, with the following interpretation:

$$\sigma \in \tau : W \xrightarrow{\langle \sigma p, \tau q \rangle} B \times \Omega^B \xrightarrow{e} \Omega$$

- Finally, we can define local functions as $\lambda$ objects, such as

$$\lambda x_C \sigma : A \to B^C$$

  where $x_C$ is a variable of type $C$ and $\sigma : C \times A \to B$.

We combine terms $\alpha, \beta$ etc. of type $\Omega$ using logical connectives $\wedge, \vee, \Rightarrow, \neg$, as well as quantifiers, to get composite terms, where each of the logical connectives is now defined over the subobject classifier $\Omega$.

- $\wedge : \Omega \times \Omega \to \Omega$ is interpreted as the *meet* operation in the partially ordered set of subobjects (given by the Heyting algebra).

- $\vee : \Omega \times \Omega \to \Omega$ is interpreted as the *join* operation in the partially ordered set of subobjects (given by the Heyting algebra). This operation gives the definition of a disjunction of two properties.

- $\Rightarrow: \Omega \times \Omega \to \Omega$ is interpreted as an adjoint functor, as defined previously for a Heyting algebra. Thus, the property of implication over SCMs is modeled as an adjoint functor.

We can combine these logical connectives with the term interpretation as arrows, relegating some details to [Mac Lane and Moerdijk, 1992]. We now turn to the Kripke-Joyal semantics of this language.

### B.3  Kripke-Joyal Semantics for a Causal Topos

We now define the Kripke-Joyal semantics for the Mitchell-Bénabou language of a causal topos. Any free variable $x$ must have some causal model $X$ of $C_\Omega$ as its type. For any causal model $M$ in $C_\Omega$, define a *generalized element* as a morphism $\alpha : N \to M$. To understand this definition, note that we can define an element of a causal model by the morphism $x : \mathbf{1} \to M$. Thus, a generalized element $\alpha : N \to M$ represents the "stage of definition" of $M$ by $N$. We specify the semantics of how an SCM $N$ supports any formula $\phi(\alpha)$, denoted by $N \Vdash \phi(\alpha)$, as follows:

$$N \Vdash \phi(\alpha) \quad \text{if and only if} \quad \text{Im } \alpha \leq \{x | \phi(x)\}$$

Stated in the form of a commutative diagram, this "forcing" relationship holds if and only if $\alpha$ factors through $\{x | \phi(x)\}$, where $x$ is a variable of type $M$ (recall that objects $M$ of a topos form its types), as shown in the following commutative diagram. [3]

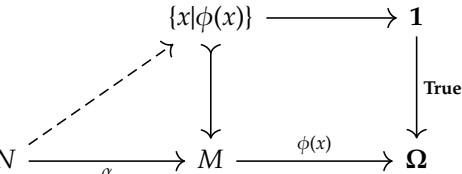

This diagram provides an interesting way to define causal interventions in a causal topos, because it defines submodels of $M$. Building on this definition, if $\alpha, \beta : N \to M$ are parallel arrows, we can give semantics to the formula $\alpha = \beta$ by the following statement:

$$N \xrightarrow{\langle \alpha, \beta \rangle} M \times M \xrightarrow{\delta_M} \Omega$$

following the definitions in the previous section for the composite $\langle \alpha, \beta \rangle$ and $\delta_X$ in the Mitchell-Bénabou language. We can extend the previous commutative diagram to show that $U \Vdash \alpha = \beta$ holds if and only if $\langle \alpha, \beta \rangle$ factors through the diagonal map $\Delta$:

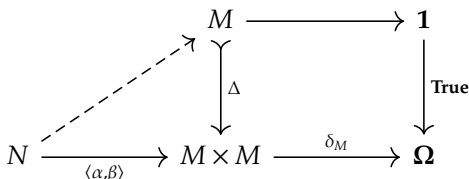

- **Monotonicity:** If $U \Vdash \phi(x)$, then we can pullback the interpretation through any arrow $f : U' \to U$ in a topos $\hat{C}$ to obtain $U' \Vdash \phi(\alpha \circ f)$.

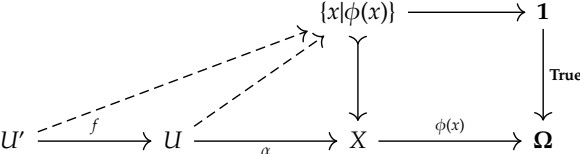

- **Local character:** Analogously, if $f : U' \to U$ is an epic arrow, then from $U' \Vdash \phi(\alpha \circ f)$, we can conclude $U \Vdash \phi(x)$.

---

[3]The concept of "forcing" is generalized from set theory [Mac Lane and Moerdijk, 1992].

**Theorem 9.** *If $\alpha : N \to M$ is a generalized element of causal model M, and $\phi(x)$ and $\psi(x)$ are formulas with a free variable x of type M, we can conclude that*

1. *$N \Vdash \phi(\alpha) \wedge \psi(\alpha)$ holds if $N \Vdash \phi(\alpha)$ and $N \Vdash \psi(\alpha)$.*

2. *$N \Vdash \phi(x) \vee \psi(x)$ holds if there are morphisms $p : O \to N$ and $q : P \to N$ such that $p + q : N + O \to M$ is an epic arrow, and $N \Vdash \phi(\alpha p)$ and $O \Vdash \phi(\alpha q)$.*

3. *$N \Vdash \phi(\alpha) \Rightarrow \psi(\alpha)$ if it holds that for any morphism $p : N \to M$, where $N \Vdash \phi(\alpha p)$, the assertion $N \Vdash \phi(\alpha p)$ also holds.*

4. *$N \Vdash \neg\phi(\alpha)$ holds if whenever the morphism $p : M \to N$ satisfies the property $N \Vdash \phi(\alpha p)$, then $N \cong \mathbf{0}$.*

5. *$M \Vdash \exists\phi(x, y)$ holds if there exists an epic arrow $p : N \to M$ and generalized elements $\beta : V \to Y$ such that $M \Vdash \phi(\alpha p, \beta)$.*

6. *$M \Vdash \forall y\phi(x, y)$ holds if for every structural causal model N, and every arrow $p : N \to M$, and every generalized element $\beta : N \to O$, it holds that $V \Vdash \phi(\alpha p, \beta)$.*

**Proof:** The proof follows readily from the general result on Kripke-Joyal semantics for the Mitchell-Bénabou languages of any topos [Mac Lane and Moerdijk, 1992] The Kripke-Joyal semantics takes on a simpler form when using a Grothendieck topology on a topos, and we postpone the details to the Supplementary Materials. □

### B.4 Kripke-Joyal Semantics for Sheaves

In the main paper, we introduced Kripke-Joyal semantics for any topos. These semantics can be specialized to a topos equipped with a Grothendieck topology, that is a site. This specialized structure captures how causal inference is woven in the fabric of the internal logic of a causal topos. Define $\mathrm{Sh}(C, \mathcal{J})$ be a topos of sheaves with a specified Grothendieck topology $\mathcal{J}$, defined by the following diagram:

$$C \xrightarrow{\text{よ}} \mathcal{P}(C) \xrightarrow{a} \mathrm{Sh}(C, \mathcal{J}) \cong C$$

where we know that the Yoneda embedding よ creates a full and faithful copy of the original category $C$. Let us define the semantics for a sheaf element $\alpha \in X(C)$, where $X(C) = \mathrm{Sh}(C, \mathcal{J})(\mathcal{C}(-, C), X)$. Since we know that $\{x|\phi(x)\}$ is a subsheaf, and given an arrow $f : D \to C$ of $C$, and $\alpha \in X(C)$, then if $\alpha$ is one of the elements that satisfies the property that $\{x|\phi(x)\}$, the monotonicity property stated above implies that $\alpha \circ f \in \{x|\phi(x)\}(D) \subseteq X(D)$. Also, the local character condition stated above implies that if $\{f_i : C_i \to C\}$ is a cover in the Grothendieck topology $\mathcal{J}$ such that $C_i| \Vdash \phi(\alpha \circ f_i)$ for all $i$, then $C \Vdash \phi(\alpha)$.

With these assumptions, we can restate the Kripke-Joyal semantics for the topos category of sheaves as follows:

1. $C \Vdash \phi(\alpha) \wedge \psi(\alpha)$ if it holds that $C \Vdash \phi(\alpha)$ and $C \Vdash \psi(\alpha)$.
2. $C \Vdash \phi(\alpha) \vee \psi(\alpha)$ if there is a covering $\{f_i : C_i \to C\}$ such that for each $i$, either $C_i \Vdash \phi(\alpha)$ or $C_i \Vdash \psi(\alpha)$.
3. $C \Vdash \phi(\alpha) \to \psi(\alpha)$ if for all $f : D \to C$, and $D \Vdash \phi(\alpha \circ f)$, it holds that $D \Vdash \psi(\alpha \circ f)$.
4. $C \Vdash \neg\phi(\alpha)$ holds if for all arrows $f : D \to C$ in $C$, if $D \Vdash \phi(\alpha \circ f)$ holds, then the empty family is a cover of $D$.
5. $C \Vdash \exists y\, \phi(x, y)$ holds if there is a covering $\{f_i : C_i \to C\}$ and elements $\beta_i \in Y(C_i)$ such that $C_i \Vdash \phi(\alpha \circ f_i, \beta_i)$ holds for each $i$.
6. Finally, for universal quantification, $C \Vdash \forall y\, \phi(x, y)$ holds if for all arrows $f : D \to C$ in the category $C$, and all $\beta \in Y(D)$, it holds that $D \Vdash \phi(\alpha \circ f, \beta)$.

## C  Local Set Theory

The Mitchell-Bénabou language is an example of a "local set theory" Bell [1988]. Formally, the Mitchell-Bénabou language for the Generalized Do-Calculus is a local set theory, defined

by a set of types that correspond to each structural causal model $M$ object in $\mathcal{C}_\Omega$. A *local set theory* [Bell, 1988] is defined as a language $\mathcal{L}$ specified by the following classes of symbols:

1. Symbols **1** and $\Omega$ representing the *unity* type and *truth-value* type symbols.

2. A collection of symbols $\mathbf{A}, \mathbf{B}, \mathbf{C}, \ldots$ called *ground type symbols*.

3. A collection of symbols $\mathbf{f}, \mathbf{g}, \mathbf{h}, \ldots$ called *function* symbols.

To instantiate this definition for our paper, the ground types will be SCMs, each of which will be interpreted as a primitive type in Section B.2. We will use the topos-theoretical constructions to construct composite types. We can use an inductive procedure to recursively construct **type symbols** of $\mathcal{L}$ as follows:

1. Symbols **1** and $\Omega$ are type symbols (the terminal object and the subobject classifier in a causal topos).

2. Any ground type symbol is a type symbol. For a causal topos, each SCM is a ground type symbol.

3. If $\mathbf{A}_1, \ldots, \mathbf{A}_n$ are type symbols, so is their product $\mathbf{A}_1 \times \ldots \mathbf{A}_n$, where for $n = 0$, the type of $\prod_{i=1}^{n} \mathbf{A}_i$ is **1**. The product $\mathbf{A}_1 \times \ldots \mathbf{A}_n$ has the *product type* symbol. These constructs allow defining an algebra of causal models.

4. If $\mathbf{A}$ is a type symbol, so is $\mathbf{PA}$. The type $\mathbf{PA}$ is called the *power* type. [4] We thus can give meaning to concept of a "powerset" of a causal model, where we interpret the subobject classifier as defining the abstract semantics of a powerset for each SCM.

Thus, a product of SCMs will define product types. Given an SCM $M$, we can define its power type as well, which is an abstract notion of the "power set" of a causal model (if you interpret this in the context of subobject classifiers, it means that we are defining a family of submodels). For each type symbol $\mathbf{A}$, the language $\mathcal{L}$ contains a set of *variables* $x_\mathbf{A}, y_\mathbf{A}, z_\mathbf{A}, \ldots$. In addition, $\mathcal{L}$ contains the distinguished $*$ symbol. Each function symbol in $\mathcal{L}$ is assigned a *signature* of the form $\mathbf{A} \to \mathbf{B}$. [5] We can define the *terms* of the local set theory language $\mathcal{L}$ recursively as follows:

- $*$ is a term of type **1**.

- for each type symbol $\mathbf{A}$, variables $x_\mathbf{A}, y_\mathbf{A}, \ldots$ are terms of type $\mathbf{A}$.

- if $\mathbf{f}$ is a function symbol with signature $\mathbf{A} \to \mathbf{B}$, and $\tau$ is a term of type $\mathbf{A}$, then $\mathbf{f}(\tau)$ is a term of type $\mathbf{B}$.

- If $\tau_1, \ldots, \tau_n$ are terms of types $\mathbf{A}_1, \ldots, \mathbf{A}_n$, then $\langle \tau_1, \ldots \tau_n \rangle$ is a term of type $\mathbf{A}_1 \times \ldots \mathbf{A}_n$, where if $n = 0$, then $\langle \tau_1, \ldots \tau_n \rangle$ is of type $*$.

- If $\tau$ is a term of type $\mathbf{A}_1 \times \mathbf{A}_n$, then for $1 \le i \le n$, $(\tau)_i$ is a term of type $\mathbf{A}_i$.

- if $\alpha$ is a term of type $\Omega$, and $x_\mathbf{A}$ is a variable of type $\mathbf{A}$, then $\{x_\mathbf{A} : \alpha\}$ is a term of type $\mathbf{PA}$.

- if $\sigma, \tau$ are terms of the same type, $\sigma = \tau$ is a term of type $\Omega$.

- if $\sigma, \tau$ are terms of the types $\mathbf{A}, \mathbf{PA}$, respectively, then $\sigma \in \tau$ is a term of type $\Omega$.

A term of type $\Omega$ is called a *formula*. The language $\mathcal{L}$ does not yet have defined any logical operations, because in a typed language, logical operations can be defined in terms of the types, as illustrated below.

- $\alpha \Leftrightarrow \beta$ is interpreted as $\alpha = \beta$.

- **true** is interpreted as $* = *$.

- $\alpha \wedge \beta$ is interpreted as $\langle \alpha, \beta \rangle = \langle \mathbf{true}, \mathbf{false} \rangle$.

- $\alpha \Rightarrow \beta$ is interpreted as $(\alpha \wedge \beta) \Leftrightarrow \alpha$

---

[4]Note that in a topos, these will be interpreted as *power objects*, generalizing the concept of power sets.

[5]In a topos, these will correspond to arrows of the category.

- $\forall x \, \alpha$ is interpreted as $\{x : \alpha\} = \{x : \textbf{true}\}$
- **false** is interpreted as $\forall \omega \, \omega$.
- $\neg \alpha$ is interpreted as $\alpha \Rightarrow \textbf{false}$.
- $\alpha \vee \beta$ is interpreted as $\forall \omega \, [(\alpha \Rightarrow \omega \wedge \beta \Rightarrow \omega) \Rightarrow \omega]$
- $\exists x \, \alpha$ is interpreted as $\forall \omega [\forall x (\alpha \Rightarrow \omega) \Rightarrow \omega]$

Finally, we have to specify the inference rules, which are given in the form of *sequents*. We will just sketch out a few, and the rest can be seen in [Bell, 1988]. A sequent is a formula $\Gamma : \alpha$ where $\alpha$ is a formula, and $\Gamma$ is a possibly empty finite set of formulae. The basic axioms include $\alpha : \alpha$ (tautology), $: x_1 = *$ (unity), a rule for forming projections of products, a rule for equality, and another for comprehension. Finally, the inference rules are given in the form:

- *Thinning:*

$$\frac{\Gamma : \alpha}{\beta, \Gamma : \alpha}$$

- *Cut*:

$$\frac{\Gamma : \alpha, \quad \alpha, \Gamma : \beta}{\Gamma : \beta}$$

- *Equivalence*:

$$\frac{\alpha, \Gamma : \beta \quad \beta, \Gamma : \alpha}{\Gamma : \alpha \Leftrightarrow \beta}$$

A full list of inference rules with examples of proofs is given in [Bell, 1988]. Now that we have the elements of a local set theory defined as shown above, we need to connect its definitions with a causal topos. That is the topic of the next section.

### C.1 Counterfactuals using Topos Categories

de Araujo Fernandes and Haeusler [2009] describe an intuitionistic logic for Lewis' [Lewis, 1973] theory of counterfactuals, where the neighborhood system of possible worlds is governed by the graph topos $C_\Gamma$ illustrated in Figure 6. However, as Galles and Pearl [1988] argued, many counterfactuals have causal meaning. For example, the well-known counterfactual "If Kangaroos had no tails, they would fall over" attains causal meaning as the logic of counterfactuals proposed by Lewis evaluates its truth in the nearest possible worlds, where the laws of physics are the same as our world, except for a "kangaroo surgery". Lewis introduces two counterfactual connectives, and we illustrate how to model one of them in our framework:

- $\alpha \,\square\!\!\rightarrow\, \beta$ is true in a world $u$ according to a neighborhood system $C_\Omega$, if either for no world $w$ in $C_\Omega(u)$, $\models_w \alpha$, or there is some neighborhood $N$ in $C_\Omega(u)$ that has a world $w$ such that $\models_w \alpha$ and $\models_v \alpha \Rightarrow \beta$ in every world $v$ of $N$.

Crucially, unlike the case for de Araujo Fernandes and Haeusler [2009], our TCM framework imposes a causal structure on the neighborhood system.

## D  Intuitionistic $j$-Do-Calculus on Sites

In a recent paper, we have developed an intuitionistic generalization of Pearl's do-calculus on sites [Mahadevan, 2025a]. A *site* $(C, \mathcal{J})$ is a small category $C$ whose objects index regimes (e.g., labs, contexts) and a Grothendieck topology $\mathcal{J}$ that specifies which families $\{u_i \rightarrow u\}$ "cover" a stage $u$. The topos $\mathbf{S}_{\mathcal{J}}(C)$ of $\mathcal{J}$-sheaves behaves like a universe of sets varying over regimes. A formula $\varphi$ is $\mathcal{J}$-stable at $u$ iff it holds on a cover of $u$ (Kripke–Joyal semantics). Intuitively: truth is verified locally and then glued. Random variables and probabilities live

internally (as objects/morphisms of $\mathbf{Sh}_{\mathcal{J}}(C)$). When $\mathcal{J}$ is trivial, these reduce to the classical notions.

Pearl's do-calculus [Pearl, 2009] is a complete axiom system for interventional identification in acyclic causal models under classical (Boolean) logic. In [Mahadevan, 2025a], we generalize it to $j$-stable causal inference inside a topos of sheaves $\mathbf{Sh}_{\mathcal{J}}(C)$, where regimes form a site $(C, \mathcal{J})$ and observations/interventions are sheaves on that site. We define $j$-stability for conditional independences and interventional claims as local truth in the internal logic of $\mathbf{Sh}_{\mathcal{J}}(C)$. We give three inference rules that mirror Pearl's insertion/deletion and action/observation exchange, and we characterize their soundness using the Kripke–Joyal semantics. In a companion paper [Mahadevan, 2025b], which is currently in preparation, we provide concrete algorithms for $j$-stable causal discovery on sites, and compare a number of popular causal discovery procedures (score-based and constraint-based) [Zanga and Stella, 2023] with their $j$-stable variants.

# E  Affine CDU and Markov Categories

In this section, we review previous work on affine CDU ("copy-delete-uniform") categories [Cho and Jacobs, 2019] and Markov categories [Fritz, 2020], which have been proposed as a unifying categorical model for causal inference, probability and statistics. They are symmetric monoidal categories, which we reviewed above, combined with a comonoidal structure on each object. Importantly, Markov categories are semi-Cartesian because they do not use uniform copying, but contain a Cartesian subcategory defined by deterministic morphisms (see below). We give a brief review of Markov categories, and significant additional details that are omitted can be found in [Cho and Jacobs, 2019, Fritz, 2020, Fritz and Klingler, 2023]. The equations are written in the diagrammatic language of string diagrams, which can be shown to represent a formal language that is equivalent to writing down algebraic equations [Selinger, 2010].

**Definition 36.** *A* Markov category *$C$ [Fritz, 2020] is a symmetric monoidal category in which every object $X \in C$ is equipped with a commutative comonoid structure given by a comultiplication* $\mathsf{copy}_X : X \to X \otimes X$ *and a counit* $\mathsf{del}_X : X \to I$, *depicted in string diagrams as*

$$\mathsf{copy}_X = \quad\quad\quad \mathsf{del}_X = \quad\quad\quad (1)$$

*and satisfying the commutative comonoid equations,*

$$\quad = \quad \quad\quad\quad (2)$$

$$\quad = \quad | \quad = \quad \quad\quad\quad\quad = \quad \quad\quad (3)$$

*as well as compatibility with the monoidal structure,*

$$\quad = \quad \quad\quad\quad\quad = \quad \quad\quad (4)$$

*and naturality of* $\mathsf{del}$*, which means that*

$$\boxed{f} = \quad\quad\quad\quad (5)$$

*for every morphism $f$.*

Let us briefly explain these definitions. The $del_X : X \rightarrow I$ is essentially like integrating over a probability distribution, which always yields 1. Hence, $I$, the unit of the tensor product, is the terminal object in affine CDU and Markov categories. Bayes rule turns into a *disintegration rule* [Cho and Jacobs, 2019], which is only available in Markov categories with conditionals (i.e., where one can categorically refine $P(y|x)$ conditional distributions). Note that in the continuous case of random variables defined as measurable functions on real numbers, one has to take considerable care in defining conditioning [Halmos, 1974]. The $copy_X$ procedure is uniform, and deterministic, meaning if you take the tensor product of two variables $X \otimes Y$ and then copy the resulting object, that's exactly the same as first copying $X$ into $X \otimes X$ and $Y$ into $Y \otimes Y$, and then taking the tensor product, along with a swap operation (see Equation 4). Only $del_X$ acts "uniformly", meaning that if you process a variable $X$ using some function $f$ and then delete $f(X)$ (meaning marginalize it), that's equivalent to simply deleting $X$. However, $copy_X$ is not defined this way, and we discuss that subtlety below, as it will be important in understanding why Markov categories are semi-Cartesian. To convert them into a topos, we need the result to be Cartesian closed, which is why we need to use the Yoneda Lemma to construct the category of presheaves to guarantee obtaining a topos.

### E.1 Cartesian Structure in Markov Categories

We now discuss a subcategory of Cartesian categories within Markov that involves uniform $copy_X$ and $del_X$ morphisms. One fundamental property of Markov categories is that they are *semi-Cartesian*, as the unit object is also a terminal object. But, a subtlety arises in how these copy and delete operators are modeled, as we discuss below.

**Definition 37.** *A symmetric monoidal category $C$ is* **Cartesian** *if the tensor product $\otimes$ is the categorical product.*

If $C$ and $\mathcal{D}$ are symmetric monoidal categories, then a functor $F : C \rightarrow \mathcal{D}$ is monoidal if the tensor product is preserved up to coherent natural isomorphisms. $F$ is strictly monoidal if all the monoidal structures are preserved exactly, including $\otimes$, unit object $I$, symmetry, associative and unit natural isomorphisms. Denote the category of symmetric monoidal categories with strict functors as arrows as **MON**. Let us review the basic definitions given by Heunen and Vicary [2019], which will give some further clarity on the Cartesian structure in affine CDU and Markov categories.

**Definition 38.** *The subcategory of comonoids* **coMON** *in the ambient category* **MON** *of all symmetric monoidal categories is defined for any specific category $C$ as a collection of "coalgebraic" objects $(X, copy_X, del_X)$, where $X$ is in $C$, and arrows defined as comonoid homomorphisms from $(X, copy_X, del_X)$ to $(Y, copy_Y, del_Y)$ that act uniformly, in the sense that if $f : X \rightarrow Y$ is any morphism in $C$, then:*

$$(f \otimes f) \circ copy_X = copy_Y \circ f$$
$$del_Y \circ f = del_X$$

Heunen and Vicary [2019] define the process of "uniform copying and deleting" in the category **coMON**, which we now relate to Markov categories. A subtle difference worth emphasizing with Definition 36 is that in Markov categories, only $del_X$ is "uniform", but not $copy_X$ in the sense defined by Heunen and Vicary [2019]. This distinction can be modeled in a cPROP category that is semi-Cartesian like Markov categories by suitably modifying the definition of the associated PROP map for copying.

**Definition 39.** *[Heunen and Vicary, 2019] A symmetric monoidal category $C$ admits* **uniform deleting** *if there is a natural transformation $e_X : X \xrightarrow{e_X} I$ for all objects in the subcategory $C_{\textbf{coMON}}$ of comonoidal objects, where $e_I = id_I$, as shown in Equation 5.*

This condition was referred to by Cho and Jacobs [2019] as a *causality* condition on the arrow $e_X$. Essentially, it states that if you process some object and then discard it, it's equivalent to discarding it without processing.

**Theorem 10.** *[Heunen and Vicary, 2019] A symmetric monoidal category $C$ has uniform deleting if and only if $I$ is terminal.*

This property holds for Markov categories, as noted in [Fritz, 2020], and a simple diagram chasing proof is given in [Heunen and Vicary, 2019].

**Definition 40.** *[Heunen and Vicary, 2019] A symmetric monoidal category C has **uniform copying** if there is a natural transformation $copy_X : X \to X \otimes X$ such that $del_I = \rho_I^{-1}$ satisfying Equation 2 and Equation 3.*

We can now state an important result proved in [Heunen and Vicary, 2019] (Theorem 4.28), which relates to the more general results shown earlier by Fox [1976].

**Theorem 11.** *[Fox, 1976, Heunen and Vicary, 2019] The following conditions are equivalent for a symmetric monoidal category C.*

- *The category C is **Cartesian** with tensor products $\otimes$ given by the categorical product and the tensor unit is given by the terminal object.*

- *The symmetric monoidal category C has **uniform copying and deleting**, and Equation 2 holds.*

As noted by Fritz [2020], not all Markov categories are Cartesian, because their $\mathbf{copy}_X$ is not uniform, but only $\mathbf{del}_X$ is. For example, consider the category **FinStoch**, where a joint distribution is specified by the morphism $\psi : I \to X \otimes Y$. In this case, the marginal distributions can be formed as the composite morphisms

$$I \xrightarrow{\psi} X \otimes Y \xrightarrow{\mathrm{del}_Y} X$$
$$I \xrightarrow{\psi} X \otimes Y \xrightarrow{\mathrm{del}_X} Y$$

But to require that in this case $\otimes$ is the categorical product implies that the marginal distributions defined as the above composites must be in bijection with the joint distribution.

## F   Social Impact

Causal inference addresses real-world problems with significant social impact, including healthcare, education, climate change, and economics. Discovering causal models has enormous potential for improving human lives in all these areas. However, our paper is largely a theoretical study of the universal properties that underlie causality. It does not address concrete algorithmic questions, and is complementary to the empirical literature in the field.

