# OpenReview forum: "Universal Causal Inference in a Topos"
_NeurIPS.cc/2025/Conference — NeurIPS 2025 spotlight_

### Official Review · Reviewer_HARx · 2025-06-22

**Clarity:** 1
**Significance:** 2
**Originality:** 2
**Rating:** 4
**Confidence:** 1

**Summary:**

Recall that a category is a "multigraph" where vertices are objects (here typically causal models) while edges/arrows are "morphisms" (here transformations of causal models, interventions). The paper starts with the definition of a "causal topos" which is this category made up of causal models. They recall the notion of topos which is a category that behaves like the category "SET" of sets. In particular, there is the notion of subobject classifier, which is the equivalent of the Boolean set {false, true}. Its existence morally says that there is a correspondence between subobjects (subsets) and characteristic functions.
The authors prove that the causal topos (or maybe a variation, not sure) is a topos (Theorem 1).
Section 3 shows that other categories based on causal models are topos (Theorem 2, 3, 4).
Section 4 recalls the notion of Heyting algebra for replacing the notion of "Boolean values".

**Questions:**

- - Is the category in Theorem 1 the same than the one in Definition 1? It seems that it is not the case because the definition is recalled in Theorem 1.
- What is the purpose of Section 4 (in simple words)
- 203 I am lost. objects are not SCM? Why are they "vertex" or "edge"?
- You say "general framework" that seems your own. But topos is not a new notion, same for many other existing notions presented in the document. What do you mean "your general framework"?

**Ethical Concerns:**

["NO or VERY MINOR ethics concerns only"]

**Final Justification:**

Thank you for the multiple answers. Please for non-specialists, please improve the writing. I have updated my score.

**Limitations:**

yes

**Paper Formatting Concerns:**

- 26 universal => \emph
- 49 I do not understand "our main contribution in this paper is to argue for...". The contribution cannot be about arguing. => What about saying clearly the result? "Our contribution is 1. to define a category CAUSAL of causal models, 2. to prove that CAUSAL is a topos". (also give a name to the category you introduce)
- between 72-73: please insert an outline of the paper because we are left lost at the end of the introduction.
- 74-94: seems to be kind of motivation. Should be in the intro as well?
- Figure 1. The left and middle parts should be drawn in the same direction. The right part is not understandable.
- 96 Remove "Thus"
- 97 subobject classifier => \emph
- 102 Very difficult to know whether Definition 1 is from the litterature or whether it is new
- 106 "right" => adequate
- 114 Definition 2 already exists in the literature. Please give a citation. Start with An elementary topos (here simply refered as topos) is... Also I think Definition of a topos should be in a background section on category theory, not after a discussion on SCM.
- Definition 1. At this time of the document, "subobject classifier" is not defined yet
- 134 add phi after "pullback" (I am not sure, if it is not phi, refer to the appropriate object or morphism)
- 136  "it is relatively easy to show that" => the reader does not the feelings of the authors here. Remove "simply" too.
- 137 (Boolean-value) function phi => according to the diagram above, the values of phi are in \Omega and are not Boolean.
- 141 x should be X?
- Theorem 1. "The category where objects blablabla..." => define the category before with a name CAUSAL (in the tradition of category theory like SET, GRP, etc.). By the way, to be sure, is it the category defined in Definition 1?
- 145 "The proof essentially involves...". Then do it. The proof have the shape. "Proof of i), proof of ii)....". The proof has not this shape. And it is not what it is done. It seems like you only tackle point (iv), but it is not said explicitely so I am not sure.

- 792 present U and V in order
- 793 f_i should be equations no? not functions
- 796 realization is not defined (the reader can guess what it is but the document should be clean)
- 798 Pearl [2009] => [Pearl 2009]
- 798 set => subset
- 822 "Our framework": what do you mean? topos already exist, SCM too.
- 834 "celebrated"???
- 839-840 "the set of all sets is not a set" => "the class of all sets is not a set"

**Quality:**

2

**Strengths And Weaknesses:**

Quality
+ The results are interesting but difficult to grasp.
- The definitions are not crystal-clean.


Clarity
- The paper is not well-organized. Some notions of category theory are in the text (subobject classifier) while others are in the appendix. It makes difficult to navigate in the document.
- The contribution is Theorem 1-4 but Section 4 for instance is very obscure.


Significance
+ The work could be of interest.
- But the paper fails showing the signifiance. We perfectly understand that there is no algorithmic contribution. That is recalled several times in the document. But the question is then: - what do we better understand now?


Originality
+ Section 2 and theorems 1-4 are new.
- Section 4 seems to only recall existing notions and the impact on the work is not explicitely stated.

---

> ### Author Rebuttal · Authors · 2025-07-28
>
> Dear Reviewer HARx:
>
> Thank you for your careful reading of the paper and your detailed comments. They will be invaluable in improving the paper. We appreciate the opportunity to address your concerns below.
>
> QUESTIONS regarding Quality:
>
> QUESTION 1: The results are interesting but difficult to grasp.
>
> ANSWER: We agree that category theory is a novel mathematical area for causal inference researchers in AI/ML, but as we mentioned in the paper, there is a recent surge of published journal papers that are showing the usefulness of categorical thinking in causality. It will take some time for ideas like topos theory to percolate into the field of causal inference in AI/ML, but we believe this will happen in due course as more researchers become familiar with the concepts. Recognizing that causal models such as SCMs form a topos category gives new insights. For example, it shows that causal interventions can be meaningfully thought about as constructing sub-objects in a category, and there is a wealth of known properties regarding sub-object classifiers in topos theory. Similarly, it introduces for the first time to our knowledge the idea of constructing "exponential objects" in causal models, so we can now precisely define what it. means to exponentiate one causal model M1 over M2, or M1^M2. The topos theoretic nature of SCM categories mandates that exponential objects must exist. For example, as we described in the paper, directed graphs form a topos, and hence they have exponential objects as well (Vigna's paper on Arxiv on the topos of graphs constructs such exponential objects, and shows their relevance to  automata models).  Similarly, we believe our work will show new ways to reason about causal models using the Mitchell-Benabou language of the SCM topos category, and this will be a topic that we aim to explore in future papers.
>
> QUESTION 2: The definitions are not crystal-clean.
>
> ANSWER: We apologize in advance for any lack of clarity, but we attempted within the confines of a conference paper to be as thorough as we could. We will of course address all your concerns in the revised paper regarding lack of clarity. We believe that each of our definitions can be rigorously stated and all our theorems can be proved rigorously, subject of course to limitations of space. Where possible, we intend to provide references to the published literature and textbooks on classic results.
>
> QUESTION 3: The paper is not well-organized. Some notions of category theory are in the text (subobject classifier) while others are in the appendix. It makes difficult to navigate in the document.
>
> ANSWER: We apologize for the deficiencies in organization and will strive to improve the organization in the revised paper. We attempted to give a concise overview of the results in the main paper, and left the more detailed definitions to the background supplementary material. In the revised paper, we will of course ensure that all key definitions are given  in the main paper.
>
> QUESTION:  The contribution is Theorem 1-4 but Section 4 for instance is very obscure.
>
> ANSWER: Section 4 describes how every topos can be characterized by an internal logical language called the Mitchell-Benabou language. It is a unique aspect of a topos as a category. For causal inference, it brings new insights because it shows that we can now use the MB language to reason about causal models. This topic of course deserves a new paper in itself, and in fact, it is an entirely new line of research that has yet to be explored in causal inference. For the revised paper, we will attempt to expand this section and give some simple examples to illustrate what the MB language could reason about in causal inference.
>
>
> QUESTIONS regarding Significance
>
> QUESTION 1: The work could be of interest.
> But the paper fails showing the significance. We perfectly understand that there is no algorithmic contribution. That is recalled several times in the document. But the question is then: - what do we better understand now?
>
> ANSWER:  To answer your valid concern here, we can clearly state what do we better understand now by mentioning the major points made in the paper. First, showing that SCMs and other causal models form a topos category means that we can now model causal interventions in terms of sub-object classifiers. We now have a new way to understand and analyze a causal intervention, something that was not known before. Second, we can give new meaning to terms that have never been explored previously, such as "exponential objects" in causal inference. We can rigorously define what it means to exponentiate one causal model M1 with respect to another, M2, so we can now talk about M1^M2, building on the property that exponential objects must exist in every topos. Third, we know now that because SCMs form a topos category, they can be studied with respect to the Mitchell-Benabou logical language and given the Kripke-Joyal semantics. This gives us a new logic to study causal inference. Finally, knowing that casual models form a topos category, we can bring in the vast power of category theory in terms of functors that map a topos category in the category of topological spaces. A recent example of this type of analysis is given in the Entropy journal paper by Mahadevan (vol 7, no. 5, 2025, "Higher Algebraic K-theory of Causal Inference), which shows that causal equivalence classes of the type studied by Chickering and Meek can be given topological meaning and we can use powerful tools from abstract homotopy theory to study the structure of causal equivalence classes. We can apply similarly many techniques from topos theory to study causal inference.
>
> QUESTIONS regarding Originality:
>
> QUESTION 1: Section 2 and theorems 1-4 are new. Section 4 seems to only recall existing notions and the impact on the work is not explicitly stated.
>
> ANSWER: As we mentioned above, Section 4 shows that each topos has an internal logic that is described by the Mitchell-Benabou language. In the revised paper, we will use a simple example to illustrate the use of the MB language to reason about causal models.
>
> QUESTION 2: Theorem 1: the category in Theorem 1 is the same as in Definition 1: objects are SCMs, and arrows are operations on SCMs, such as interventions. We begin with this theorem since its proof is simpler than the following theorems (3 and 4).
>
> ANSWER: Theorems 3 and 4 generalize Theorem 1, since they apply to a larger class of causal models (Markov categories and simplicial sets), and use a more powerful set of theoretical tools (Yoneda embeddings, sheaves over a topos), whereas the proof of Theorem 1 is much simpler in comparison.
>
> We are indebted to you for your careful reading of the paper, and your detailed comments regarding typos. We will of course make the necessary revisions to address all your concerns.  A few highlighted comments:
>
> QUESTION: 49 I do not understand "our main contribution in this paper is to argue for...". The contribution cannot be about arguing. => What about saying clearly the result? "Our contribution is 1. to define a category CAUSAL of causal models, 2. to prove that CAUSAL is a topos". (also give a name to the category you introduce)
>
> ANSWER: We absolutely agree with you and will revise the paper accordingly.
>
> QUESTION: between 72-73: please insert an outline of the paper because we are left lost at the end of the introduction.
>
> ANSWER: Yes, thank you, we will include the outline as requested.
>
> 74-94: seems to be kind of motivation. Should be in the intro as well?
>
> ANSWER: Yes, we will include it, thank you.
>
> QUESTION regarding Figure 1. The left and middle parts should be drawn in the same direction. The right part is not understandable.
>
> ANSWER: The right part of  Figure 1 describes the construction of the sub-object classifier for the SCM category. We will revise the figure to make it clearer.
>
> QUESTION regarding 102 Very difficult to know whether Definition 1 is from the litterature or whether it is new
>
> ANSWER: We will ensure that all standard definitions are cited.
>
> QUESTION regarding 134 add phi after "pullback" (I am not sure, if it is not phi, refer to the appropriate object or morphism)
>
> ANSWER: Pullback refers to the whole diagram. We will clarify in the revised paper.
>
> QUESTION about Theorem 1. "The category where objects blablabla..." => define the category before with a name CAUSAL (in the tradition of category theory like SET, GRP, etc.). By the way, to be sure, is it the category defined in Definition 1?
>
> ANSWER: Yes, Theorem 1 is for the category defined in Definition 1.
>
> QUESTION about 145 "The proof essentially involves...". Then do it. The proof have the shape. "Proof of i), proof of ii)....". The proof has not this shape. And it is not what it is done. It seems like you only tackle point (iv), but it is not said explicitely so I am not sure.
>
> ANSWER: We only wrote the proof for the sub-object classifier, and left the rest to the Supplementary Materials. We will revise and condense the proof in the revised paper.
>
> QUESTION about 822 "Our framework": what do you mean? topos already exist, SCM too.
>
> ANSWER: By "Our framework", we mean treating the SCM category as a topos, which is our novel contribution, to the best of our knowledge.
>
> Thank you again for your careful reading of the paper, and for your many excellent comments regarding the presentation and its clarity. We will of course strive to incorporate all your suggestions in the revised paper.

---

> > ### Comment · Reviewer_HARx · 2025-08-01
> > **Response to the rebuttal**
> >
> > Thank you for the detailed rebuttal.
> >
> > You say that "exponential objects" exist. Could you say more about the impact of the existence of exponential objects? What does it mean (in simple words, for persons that are not expert in category theory)? Otherwise, people outside from category theory will just say "OK, and so what?" or even worse: "OK, I do not care".
> >
> > I understand that it is difficult sometimes to convey your ideas in a conference paper. But still I am happy that you agree that to improve the organisation and the presentation. Due to the general audience of the conference, it is important that your ideas are motivated, and properly explained. Your proposal "we will use a simple example to illustrate the use of the MB language to reason about causal models." is certainly a good idea.

---

> > > ### Author Response · Authors · 2025-08-03
> > > **Exponential objects in causal inference**
> > >
> > > Dear Reviewer HARx:
> > >
> > > Thank you again for your comments. Your question about the usefulness of exponential objects in causal inference is an excellent one, and we would like to briefly explain how they can be applied to model causal equivalences and causal interventions. As a reminder, the category ${\cal C_\Omega}$ was defined in the paper as one whose objects are structural causal models (SCMs) and whose arrows are operations on SCMs, such as causal interventions or edge reversals etc, which are used extensively in causal discovery. To define exponential objects, let us model each SCM M by its unique function $f: U \rightarrow V$, mapping exogenous variables to endogenous variables. Thus, we can think of the category ${\cal C_\Omega}$ as consisting of objects defined as functions $f: U \rightarrow V$. The arrows of ${\cal C_\Omega}$ are commutative diagram as shown on line 153 in the paper. Each arrow in ${\cal C}_\Omega$ is one of these commutative diagrams, where in the diagram, $f,f'$ denote two SCMs M and M', and $g$ and $h$ are functions on endogenous and exogenous variables that make the diagram commute (i.e., we must assert that $g \circ f = f' \circ h$).
> > >
> > > Now, given two SCMs M and M', defined by their associated functions $f: U \rightarrow V$ and $f': U' \rightarrow V'$, we define the exponential object $f^{f'}: E \rightarrow V'^{V}$, where $E =  \( \langle g, h \rangle | g \circ f = f' \circ h \)$. That is, $E$ is the set of all functions $g, h$ in the commutative diagram shown on line 153 in the paper that makes the diagram commute. Also, $V'^V$ is simply the exponential object in the category of Sets, that is, all functions $t: V \rightarrow V'$. What does this definition mean? It tells us that the exponential object is defined on the domain of the set of all operations that turns an SCM model M into another SCM model M', and that because arrows compose, these commutative diagrams can be stacked horizontally.
> > >
> > > Now, let us turn to describe the use of such exponential objects in causal discovery. Chickering in his JMLR paper on "Optimal Structure Identification using Greedy Equivalent Search" (JMLR: https://www.jmlr.org/papers/volume3/chickering02b/chickering02b.pdf) described the GES algorithm for causal discovery, where the set of equivalence classes of DAGs with respect to some fixed observational data set was modeled through a process of edge reversals of "covered" edges. Chickering proved the conjecture of Meek that the space of all causal DAG models could be traversed by edge reversals, along with edge deletion and insertion. We can use the notion of exponential objects described above to define a variant of Chickering's GES procedure, where the equivalence class generated by edge reversal would be modeled as compositions of commutative diagram arrows of the type shown on line 153 in our paper. We can also define the meaning of the exponential of one SCM DAG M with respect to another SCM DAG M' in terms of causal equivalence classes.
> > >
> > > A second application is to causal intervention. As we defined in the paper, causal interventions correspond to (non-Boolean) subobject classifiers in the topos defined by the category of SCM models ${\cal C_\Omega}$. The particular commutative diagram shown on line 153 that corresponds to a causal intervention would be required to use ``monic"  $g,h$  (monic means injection, because in an intervention, the set of exogenous or endogenous variables permitted would be a subset of those in the non-intervened model). Now, the exponential object would be defined over the space of all operations $g,h$ on endogenous and exogenous variables that are the result of a ${\bf do}$-calculus intervention as defined by Pearl.
> > >
> > > Finally, in this paper we did not explicitly address causal "homotopy", meaning the property that non-isomorphic objects may in fact be required to be treated as if they are isomorphic. In causal discovery, we know that from data, we may not be able to tell one causal model from another. This fact was the basis of Chickering and Meek's work on edge reversals. Meek's edge reorientation rules are widely used in many causal discovery algorithms, such as FCI and its many variants. To make the connection more precise, note that equivalent causal models in the category ${\cal C_\Omega}$ correspond to non-isomorphic objects that are actually two different causal models that are observationally indistinguishable. There is a rich literature in categorical homotopy theory that shows how to construct the homotopy category where such non-isomorphic objects are treated as if they are isomorphic. We can apply this theory to the topos category ${\cal C_\Omega}$ to model the process of treating causally equivalent models as if they are equal. In Chickering's GES algorithm maintains for each causal equivalence class one sample model as the exemplar., creating in effect a homotopy category.
> > >
> > > We thank you for giving us this opportunity to explain our work further.

---

> > > > ### Comment · Area_Chair_HWTp · 2025-08-06
> > > > **Feedback**
> > > >
> > > > Dear HARx, would you be that kind to share your opinion on the repky from the authors?

---

> > > > ### Comment · Reviewer_HARx · 2025-08-07
> > > > **Response about exponential objects**
> > > >
> > > > $V'^V$ is simply the exponential object in the category of Sets.
> > > > Do we also have $V'^V$ to be exponential object in the category ${\cal C_\Omega}$?
> > > >
> > > > Sorry to be a bit pragmatic. $V'^V$ is just the powerset of $V'$ by $V$ (for the set of functions from $V$ into $V'$). So why would we need category theory to construct/define $V'^V$?

---

> > > > > ### Author Response · Authors · 2025-08-07
> > > > > **Exponential objects in causal categories**
> > > > >
> > > > > Dear Reviewer HARx:
> > > > >
> > > > > We are not using category theory to define $V'^V$, which as we remarked is indeed the exponential object in the category of sets, but we are using category theory to define the exponential objects over SCMs (which induce unique functions from the exogenous variables $U$ to the endogenous variables $V$). There is nothing in set theory that tells us what such an exponential object should be. We need category theory to help us define such exponential objects (just as category theory tells us how to define exponential objects over graphs, which also form a topos).  Of course, as in all such definitions in mathematics, we eventually boil down a new concept in terms of concepts we already understand, so the exponential object for the category of SCMs is defined eventually in terms of the exponential object in the category of sets. The reason we can do this reduction is precisely because Judea Pearl defined SCMs as inducing a unique set-valued function from exogenous to endogenous variables (one that is in fact deterministic).
> > > > >
> > > > > So, to clarify and make sure we are on the same page, the primary objects in the category of SCMs ${\cal C_\Omega}$ are SCMs defined as functions $f: U \rightarrow V$. Given two such objects, say $f_1$ and $f_2$ representing two SCMs, since this category defines a topos,  we want to now define the exponential object $f_1^{f_2}$. This is not a set-theoretic concept. We have to define what this means, which we explained how to do in our rebuttal. One has to construct the set of arrows between two SCMs, which are commutative diagrams as illustrated on line 153. Then, one defines the exponential object in the SCM category as having its domain the set of all possible arrows between two SCMs, and having its co-domain be the set of all functions $V'^V$.
> > > > >
> > > > > In the more general case, we cannot rely on this approach because SCMs are indeed very special. Hence, as we described in the paper,  we construct the topos category from Yoneda embeddings of more general causal categories, such as Markov categories, and the exponential object must be constructed in a different way than we were able to do for SCMs. Let ${\cal C}$ denotes a Markov category (which is a symmetric monoidal category with a comonoidal `"copy-delete" operation defined on each object. Fritz (https://arxiv.org/abs/1908.07021) wrote a 100-page journal paper showing how Markov categories provide a unifying categorical framework for probability and statistics, showing how many interesting theorems from d-separation to sufficient statistics can all be formulated using Markov categories. To construct exponential objects in a Markov category, we first "toposify" the category by constructing the Yoneda embeddings, where each object $c$ is mapped to a contravariant functor ${\cal C}(-, c)$.  This is simply the set of all morphisms that map into object $c$.  A remarkable result in category theory (Yoneda Lemma) shows that all objects are definable in this way up to isomorphism (so, objects essentially are defined by interactions or arrows between them).
> > > > >
> > > > > A very general result in topos theory shows that such Yoneda embeddings create a full and faithful copy of the category, and we can construct exponential objects following the standard approach described in the book hy Maclane and Moerdijk ("Sheaves in Geometry and Logic: A First Introduction to Topos Theory", Springer).  Section 1.6, page 44, of this book describes how to construct exponential objects of Yoneda embeddings. Also, Proposition 1 in Chapter 1 in this book states the following result:
> > > > >
> > > > > Proposition 1: For any (small) category, the functor category ${\cal C}^{op}$ is Cartesian closed. (page 46 in the Maclane-Moerdijk book).
> > > > >
> > > > > A "functor category" is a category where each object is a functor, and the arrows are natural transformations from one functor to another. Each object in the functor category ${\cal C}^{op}$ is a Yoneda embedding, say ${\cal C}(-, c)$. To understand why Yoneda embeddings create a functor category, note that we are mapping any object $c$ into the set ${\cal C}(-, c)$. This means we are letting the first argument "float" and be replaced by any object in the category. What this general result tells us is that Yoneda embeddings construct Cartesian closed categories, which have the property of possessing exponential objects $c^d$ for any two objects $c$ and $d$ in the category. By way of explanation, a "small" category is one with a set's worth of objects (there are "large" categories where one has to deal with proper classes of objects).
> > > > >
> > > > > I hope this clarifies the use of exponential objects in the paper. We need category theory to construct these, since it is not obvious how to define exponential objects over SCMs, or indeed over (directed) graphs. Thank you again for your question.

---

> > > > > > ### Comment · Reviewer_HARx · 2025-08-08
> > > > > > **Thank you**
> > > > > >
> > > > > > Thank you for your answer. I understand a bit better now. Sorry for the fact that it was not clear from your first answer.

---

### Official Review · Reviewer_EdR2 · 2025-06-28

**Clarity:** 3
**Significance:** 1
**Originality:** 3
**Rating:** 3
**Confidence:** 1

**Summary:**

This paper proposes a unified framework for causal inference based on topos theory, aiming to reveal the universal properties of causal inference from a category theory perspective. The authors argue that topos, as a category type with "set-like" properties, provides three key universal properties for causal inference: first, a general theory based on sheaves theory in topos that can combine local functions defining "independent causal mechanisms" into consistent global functions; second, using subobject classifiers in topos categories to provide a universal method for defining causal interventions; finally, generating an internal logical language for causal and counterfactual reasoning from the topos itself.

**Questions:**

What real-world causal inference problems are better suited for this framework compared to existing frameworks? The paper does not seem to mention how this framework improves upon existing causal inference methods.

**Ethical Concerns:**

["NO or VERY MINOR ethics concerns only"]

**Limitations:**

In my personal view, causal inference exists primarily as a tool for scientific discovery. This paper conducts excessive mathematical formalization of existing causal frameworks, which may not reflect the demands of real-world scenarios.

To AC: I have insufficient knowledge of the topos field, perhaps the AC needs to seek an emergency reviewer.

**Quality:**

3

**Strengths And Weaknesses:**

Strength:
1. This paper reduces the gaps in past causal inference work by providing a unified mathematical model for different causal representations from a topos perspective;
2. The proof process is rigorous, definitions are clear, forming a relatively complete theoretical system;
3. The concept is novel, and related work indicates that there have been no similar attempts in the past.

Weakness
1. The theory is quite complex, lacks useful scenarios, and has a high barrier for causal inference researchers;
2.  Insufficient motivation, fails to adequately explain the necessity of using a unified framework.

---

### Official Review · Reviewer_H13Y · 2025-07-01

**Clarity:** 4
**Significance:** 4
**Originality:** 4
**Rating:** 6
**Confidence:** 4

**Summary:**

The authors take a novel look at causal inference models, most notably Structural Causal Models (SCMs), and reformulate them as a topos. With this they aim to uncover a deeper understanding of causality.
I find this deeply interesting, because different Definitions/representations of casualty do vary, but seem connected on some level. It looks to me like topoi might be a good lense through which to view these connections. I love this paper, because the idea seems obvious once you read it, but very hard to spot without. As such I am curious to see what this discovery will yield in the future.

**Questions:**

Q1: Would it be possible (within page limits...) to briefly connect to the philosophy of causality literature, e.g. Sander Beckers work:
Beckers, S. (2021). Causal sufficiency and actual causation. Journal of Philosophical Logic, 50(6), 1341-1374.
(A 'No' is a perfectly fine answer!)

Q2:  I believe this work to have great impact on the future understanding of causality. Can you perhaps add a brief outlook on what viewing causality as a topos might allow us to see? Maybe some space can be gained by omitting the pointer that this paper is not for practioners and/or causal discovery.

**Ethical Concerns:**

["NO or VERY MINOR ethics concerns only"]

**Final Justification:**

As already written in the comment section:
In the causality community we only just begin to understand what causality is or what notion of causality works best. Pearl's notion is very influential, but by far not the only contender. In my view this paper could be a key insight to move this whole research line further. I agree that the paper can be improved, but in my view most papers can and this paper really has a novel contribution with -in my expectation- big impact down the road. I think that this paper deserves the visibility NeurIPS provides and I would expect it to have some strong impact down the road.

**Limitations:**

yes

**Paper Formatting Concerns:**

the formatting seems very good; especially the graphs are wonderful.

**Quality:**

4

**Strengths And Weaknesses:**

Overall: I love the core idea and think it will have wide ranging impact on our understanding of causality - especially different notions of it. I think this paper is a great foundational frame to find similarities and analogies between definitions and come closer to a 'universal' understanding of causality.

Strength:
- Great idea.
- The text is very well structured and all relevant background is explained in the appendices.
- The math - as far as I can fully understand it in my available time- seems to check out fine.

Weakness:
- Like any deep paper it would benefit from more available space...
- I would find it interesting to connect this work with Sander Beckers' work on causal sufficiency and different definitions of casualty:
Beckers, S. (2021). Causal sufficiency and actual causation. Journal of Philosophical Logic, 50(6), 1341-1374.
- In general, this paper would benefit from closer ties to the philosophy part of the causality literature - but I totally understand that space here is too limited for that.

---

### Official Review · Reviewer_giku · 2025-07-02

**Clarity:** 3
**Significance:** 3
**Originality:** 3
**Rating:** 5
**Confidence:** 4

**Summary:**

This paper approaches causal inference using the language of category
theory, and shows that several causal models (e.g. structural causal
models and bayesian networks) can be interpreted as toposes, a
specific kind of category. The paper claims that the benefit of the
interpretation is to have access to three properties: 1) the
possibility to reason about global functions starting from local
functions using sheaves in a topos, 2) the possibility of reasoning
about causal interventions using an intuitionistic logic which emerges
from their definition using subobject classifiers, 3) the possibility
of using the internal language of a topos for causal and
counterfactual reasoning.

**Questions:**

- Can you elaborate on the concrete benefit you see for a causal inference scholar to using the framework you propose?

- I didn't get why in causal models the subobject classifier is not boolean. I think I see this intuitively, but now where it comes from technically. Can you elaborate on it?

**Ethical Concerns:**

["NO or VERY MINOR ethics concerns only"]

**Limitations:**

yes

**Quality:**

3

**Strengths And Weaknesses:**

Strengths

- the paper shows that topos have the natural mathematical structure
  needed to interpret several causal model

- the paper identifies three benefits of interpreting causal models in
  toposes

- the paper does a good job of explaining abstract categorical
  concepts and their connection to causality

Weaknesses

- the paper explains well the properties that one can get from using
  toposes to interpret causal models but does not explain what one gains
  in doing this

- Some of the proofs are less rigorous than what one could expect


This is a rather unorthodox neurips submission aiming at using
abstract mathematical concepts from category theory to explain causal
inference models. This is part of a trend that has emerged in recent
years and that has brought to interpretations of several probabilistic
and statistical machine learning concepts in terms of categorical
concepts.

The main observation that the paper makes, i.e. that causal models can
be interpreted in toposes, is interesting since toposes are a
well-studied kind of category which emerges naturally in several
applications. The paper shows several benefits, from the modeling
perspective, of doing this. However, I find that the paper stops too
early and does not clarify how these modeling benefits can be really
translated to the understanding of causal models. In other terms, I
find that the paper does not explain enough what one gains from having
the three universal properties that the paper focuses on, and why a
causal inference scholar should care about them. I think this is
important for a paper of this kind, to be really appreciated.

The presentation of the paper is good. The paper introduces the
different abstract concepts in an intuitive way and connects them well
with causal inference. As a result, I think this paper could result
accessible also for reader without a deep background in category
theory. However, a downside of this is that the proofs in the paper
are less rigorous than the usual in this area. The theorem themselves
are not very deep, they are just showing the correctness of the
different interpretations, so I think it is fine to have more informal
proofs but I was a bit surprised at first.

Overall I think this is a good paper but it would be even better if
the authors could provide more explanation of what one gains from
the causal inference perspective by understanding the paper results.

---

### Decision · Program_Chairs · 2025-09-17

**Decision:**

Accept (spotlight)

**Comment:**

This paper explores the use of topos theory, a sophisticated concept from category theory, as a foundational framework for interpreting causal models. It contributes to a growing trend in machine learning that seeks to recast probabilistic and statistical concepts in abstract mathematical terms. The central claim is that toposes naturally accommodate the structure needed to represent various causal models, and the paper identifies three key benefits of doing so. It also succeeds in making these abstract ideas accessible, offering intuitive explanations that connect categorical concepts to causal inference, which could help broaden understanding among researchers unfamiliar with category theory.
However, the paper’s impact is limited by several factors. While it effectively outlines the properties gained by interpreting causal models within toposes, it does not convincingly explain why these properties matter from a causal inference perspective. The practical implications of the universal properties it highlights remain vague, and the paper does not clearly articulate what a causal inference researcher gains by adopting this framework. This lack of motivation weakens the case for its relevance, especially for readers seeking concrete benefits or applications.
The mathematical rigor is also uneven. Although the proofs are generally correct and serve to validate the interpretations, they are less formal than expected in this domain. Some reviewers find this acceptable given the paper’s conceptual nature, but others note that the lack of depth in the theorems and the informal presentation may detract from its credibility in more mathematically rigorous circles.
The organization of the paper is another point of critique. Key concepts are scattered between the main text and appendices, making it difficult to follow. Some sections, particularly Section 4, are described as obscure, and the definitions are not always presented with the clarity needed for such a complex topic. Moreover, while the paper emphasizes its theoretical contribution, it does not sufficiently connect its ideas to existing philosophical literature on causality, which could have strengthened its conceptual grounding.
Despite these weaknesses, the paper is recognized for its originality. The use of topos theory in causal inference is novel, and the foundational perspective it offers could pave the way for deeper insights into the nature of causality. However, to fully realize its potential, the paper needs to better articulate the significance of its results, clarify its methodology, and connect more explicitly to both practical scenarios and philosophical discussions.
The discussion was intense and with several posts from authors and reviewers. This helped some reviewers to re-evaluate her/his first rating of the manuscript and in particular to increase the final rating.